microbiology/biomechanics/bioengineering

fungi, microfluidics, growth

**Authors for correspondence:**
Antoine Fayeulle
e-mail: antoine.fayeulle@utc.fr
Anne Le Goff
e-mail: anne.le-goff@utc.fr

# Microfluidic monitoring of the growth of individual hyphae in confined environments

Claire Baranger[1], Antoine Fayeulle[1] and Anne Le Goff[2]

[1]Université de technologie de Compiègne, ESCOM, TIMR (Integrated Transformations of Renewable Matter), and [2]Université de technologie de Compiègne, CNRS, Biomechanics and Bioengineering, Centre de recherche Royallieu - CS 60 319 - 60 203 Compiègne Cedex, France

 AF, 0000-0001-7760-1902; ALG, 0000-0001-8965-4892

Soil fungi have the ability to form large mycelial networks. They rely on the resources available in the soil to produce biomass and are able to degrade complex biomolecules. Some of them can even degrade recalcitrant organic pollutants and are considered as promising candidates for soil bioremediation strategies. However, the success of this approach depends on the ability of fungi to colonize the soil matrix, where they encounter spatial and temporal variations of confinement, humidity and nutrient concentration. In this paper, we present a study of fungal growth at the scale of single hyphae in a microfluidic device, allowing fine control of nutrient and water supply. Time-lapse microscopy allowed simultaneous monitoring of the growth of dozens of hyphae of *Talaromyces helicus*, a soil isolate, and of the model fungus *Neurospora crassa* through parallel microchannels. The distributions of growth velocity obtained for each strain were compared with measurements obtained in macroscopic solid culture. For the two strains used in the study, confinement caused the growth velocity to drop in comparison with unconfined experiments. In addition, *N. crassa* was also limited in its growth by the nutrient supply, while the microfluidic culture conditions seemed better suited for *T. helicus*. Qualitative observations of fungi growing in microfluidic chambers without lateral confinement also revealed that side walls influence the branching behaviour of hyphae. This study is one of the first to consider the confinement degree within soil microporosities as a key factor of fungal growth, and to address its effect, along with physicochemical parameters, on soil colonization, notably for bioremediation purposes.

## 1. Introduction

Filamentous fungi play a key role in terrestrial ecosystems, through their participation in nutrient cycling, through their interactions with other organisms and a structural component of

soil [1]. They are non-motile organisms that colonize their environment through hyphal growth, thereby forming a three-dimensional mycelial network. The filamentous form allows them to grow in confined environments and colonize porous solid matrices, such as the soil or plant tissues. Hyphae grow from the tip and are able to branch either laterally or apically. This morphology is adapted to their heterotrophic mode of nutrition, enabling fungi to scout their surroundings and absorb nutrients at the growing tips of hyphae. Moreover, hyphae are able to break air-water interfaces, thus crossing air pockets and gaps between soil particles. Their morphology combined with biodegradation abilities makes filamentous micromycetes particularly interesting for soil bioremediation. In particular, some organic pollutants of concern can be used as carbon and energy source by strains that are able to access and degrade these molecules. Besides, the enzymatic diversity developed by fungi to specialize on substrates poorly degraded by competing bacteria makes them suitable for the bioremediation of recalcitrant pollutants such as polycyclic aromatic hydrocarbons (PAH) [2].

In this context, a soil-borne fungal strain, *Talaromyces helicus*, has been selected as particularly efficient for the biodegradation of PAH both in mineral medium and in historically contaminated industrial soils [3]. In order to extrapolate the efficiency of soil bioremediation by this strain over time on a given soil volume, it is of particular interest to study the growth of this strain in microenvironments approaching the scale of soil microporosities. However, the methods traditionally used for fungal culture in the lab, i.e. solid agar plates or liquid medium in shake flasks, do not reproduce the heterogeneous and compartmentalized microhabitat of fungi at the structural and functional levels.

Natural soils are porous media with high heterogeneity in pore size and distribution and can be subjected to variations in water saturation and infiltration rates depending on weather conditions. As a result, average infiltration rates through a given volume of soil can vary in the range of a few millimetres to a few tens of centimetres per hour, while local flow velocities vary greatly depending on pore size and saturation, reaching up to $50 \, \mathrm{cm \, s^{-1}}$ in cm-wide macropores [4]. Transfer of nutrients, metabolites, genes and viruses typically occurs at the submillimetric scale of soil aggregates and cannot be captured in controlled homogeneous model environments with little to no confinement [5].

Microfluidic tools have been proposed to study the interactions between microorganisms and their microenvironment, specifically in soils [6,7]. Numerous set-ups have been designed for bacteria [8], but devices intended for fungi, and more specifically filamentous fungi, remain scarce. Over the past 15 years, several studies have used fungal culture in microfluidic devices: to monitor fungal growth patterns in maze-like structures [9], highlight the fungicidal activity of *Bacillus subtilis* on the fungus *Coprinopsis cinerea* [10], investigate circadian cycles in *Neurospora crassa* [11] or image spore germination of *Fusarium fujikuroi* [12]. By using micropatterned surfaces to probe the growth of *Candida albicans* in filamentous form, geometric constraints have been shown to affect hyphal growth and morphology [13].

Most of the studies of fungal growth in microfluidic devices were performed in static conditions, i.e. in a resting fluid or gel culture medium. In such a system, the dead volume in the chamber is small and the growing microorganisms are likely to quickly consume the nutrients and oxygen initially present in the device. For animal cell culture, dynamic cell culture systems have been developed to avoid this risk of starvation [14,15].

In this study, we propose a microfluidic device to monitor the growth of individual hyphae experiencing lateral confinement in microchannels. This device is suited for microscopic observations and nutrient perfusion. Observations can be performed in static or dynamic conditions. The objective is to use these results to help discriminate between phenomena that result from the geometrical constraints and those that are merely a consequence of starvation.

The growth of the two fungal strains *Talaromyces helicus* and *Neurospora crassa* was studied with and without the implementation of a medium perfusion in the microfluidic chamber and compared with macroscopic growth on solid medium. We also studied more qualitatively the branching events that took place in the microchannels and the perfusion chamber. Apical extension velocity and branching are two key features that influence the ability for a fungus to colonize a soil.

# 2. Experimental section

## 2.1. Fungal strains and macro-culture conditions

A strain of the filamentous fungus *Talaromyces helicus* from our laboratory collection and previously isolated from an industrial contaminated soil was used for this study. The model species *Neurospora crassa* was obtained from the BCCM/MUCL Agro-food & Environmental Fungal Collection (strain

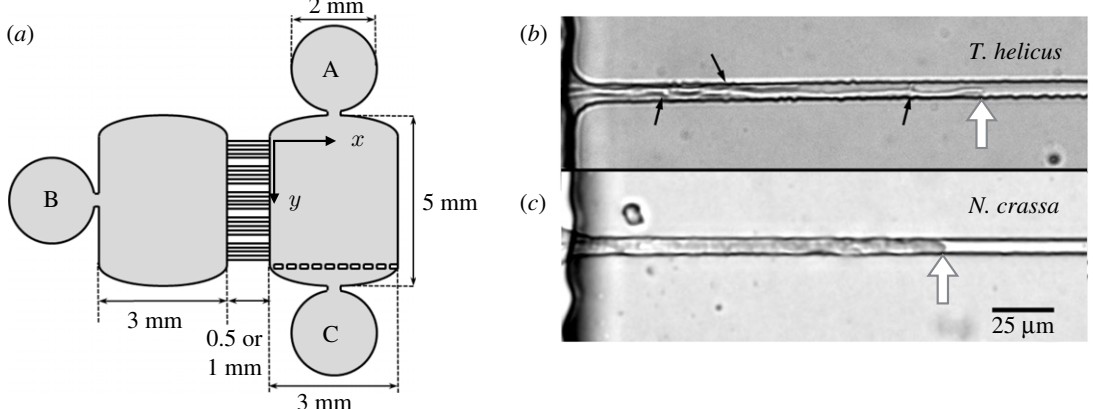

**Figure 1.** (*a*) Diagram of the microchip pattern. Microchannels for confined hyphal growth are oriented in the *x*-direction and the main chamber is perfused in the *y*-direction. A and B represent the perfusion and inoculation inlets, respectively, while C represents the perfusion outlet. Detail of (*b*) *T. helicus* and (*c*) *N. crassa* hyphae growing through 10 μm wide and 5.8 μm deep microchannels with a rectangular cross-section. The hyphal tip and branching points are indicated with white and black arrows, respectively.

MUCL 041473). Strains were maintained on MYEA medium composed of malt extract $20 \, \text{g l}^{-1}$, yeast extract $2 \, \text{g l}^{-1}$ and agar $15 \, \text{g l}^{-1}$, at 22°C with a 12 h–12 h light–dark cycle, and transplanted onto fresh medium every 10 days.

## 2.2. Microfabrication

A design, represented in figure 1*a*, featuring two chambers connected through 50 parallel microchannels was patterned onto a silicon plate by photolithography, yielding a positive master (Microfactory, Paris). Each channel is 10 μm wide and 5.8 μm deep, in order to accommodate a single hypha, as illustrated by figure 1*b,c*. Microchannels are 0.5 or 1 mm long, which allows to observe full microchannels in the microscope field, using either a ×20 or ×10 lens. The chambers were 124 μm high. Polydimethylsiloxane (PDMS, Sylgard 184, Dow Corning) was cast against the silicon wafer and allowed to cure for at least 2h at 70°C. Inlets were added by punching holes through the PDMS with a 1 mm (perfusion inlets) or 2 mm wide (inoculation inlet) biopsy puncher in the 2 mm wide circular areas visible in figure 1*a*. The negatively patterned PDMS slab was bound to a glass microscope slide after surface oxidization in a plasma chamber (Harrick) for 60 s.

## 2.3. Characterization of fluid and mass transfer in the system

In order to characterize solute transport in the device, a chip featuring 1 mm long channels was filled with distilled water, then perfused with a $50 \, \text{mg l}^{-1}$ fluorescein solution at $9 \, \mu\text{l min}^{-1}$ through the distal chamber using a peristaltic pump (Ismatec). The intensity of fluorescence in the chambers was monitored by fluorescence microscopy over 3 h using the fluorescein isothiocyanate (FITC) filter. Average fluorescence intensity at five different *y* positions in the perfusion chamber was measured for each time point. The mean grey value was measured using ImageJ software for each image over a $130 \times 600 \, \mu\text{m}$ rectangular area as close as possible to the channel openings (but not including them).

## 2.4. On-chip fungal culture

Mineral medium supplemented with glucose (MMG) at pH 5.5, as described by Fayeulle *et al.* [3], was used for all on-chip cultures: it is composed of KCl $0.25 \, \text{g l}^{-1}$, $NaH_2PO_4 \cdot 2H_2O$ $1.54 \, \text{g l}^{-1}$, $Na_2HPO_4$ $8 \, \text{mg l}^{-1}$, $MgSO_4 \cdot 7H_2O$ $0.25 \, \text{g l}^{-1}$, $NH_4NO_3$ $1 \, \text{g l}^{-1}$, $ZnSO_4 \cdot 7H_2O$ $1 \, \text{mg l}^{-1}$, $MnCl_2 \cdot H_2O$ $0.1 \, \text{mg l}^{-1}$, $FeSO_4 \cdot 7H_2O$ $1 \, \text{mg l}^{-1}$, $CuSO_4 \cdot 5H_2O$ $0.5 \, \text{mg l}^{-1}$, $CaCl_2 \cdot 2H_2O$ $0.1 \, \text{mg l}^{-1}$, $MoO_3$ $0.2 \, \text{mg l}^{-1}$ and glucose $20 \, \text{g l}^{-1}$. All chips were filled with sterile MMG prior to seeding with mycelium. A 2 mm diameter mycelium plug was collected from a solid-medium culture of *T. helicus* or *N. crassa* at the edge of the growing colony and transferred to the inoculation inlet of the chip. The inlet was then closed with a PDMS plug to prevent drying and the system was placed in a sealed Petri dish in a water-saturated atmosphere. The mycelium was allowed to grow at 22°C with a 12 h–12 h light–dark cycle, until

hyphae reached the opening of the channels. After pre-incubation, systems were placed on the stage of the microscope. Systems used in dynamic conditions were connected to a peristaltic pump (Ismatec) and the distal chamber was perfused with MMG at 1.8 µl min⁻¹, using a PTFE tubing (0.56 µm I.D. and 1.07 µm O.D.) directly plugged in the PDMS and connected to the silicon peristaltic tubing (Tygon, 0.13 mm I.D and 1.82 mm O.D.). The experimental set-up is pictured in electronic supplementary material, figure B1. Systems used in static conditions were not perfused. Hyphal growth was monitored in parallel in multiple microchannels over 10 h. The experiment was repeated three times for each strain, with and without perfusion with MMG. Growth was measured in a total of 280 channels from 15 separate chips, 6 in static conditions and 9 in dynamic conditions.

## 2.5. Imaging and data analysis

In order to measure colony growth rates on a solid medium, *N. crassa* and *T. helicus* were each inoculated in three Petri dishes on MYEA and incubated at 22°C. Pictures of the *N. crassa* colonies were taken every 6 h for 48 h. For *T. helicus*, pictures were taken every day for 4 days, and then every 2 or 3 days until colonies reached the edge of the dish. All image analysis was performed in ImageJ. Colony size on each image was measured manually due to the low contrast. For each image, the maximum and minimum diameters of the colony were measured, and the mean of both values was calculated. Microscopic observations were made using an inverted microscope (Leica DMI-8) equipped with a motorized stage, and images of five fields for each chip were taken at a 10 × magnification (or 4 × for the fluorescein perfusion) with a camera (Leica DFC 3000G), every 10 min for *T. helicus* and every 5 min for *N. crassa*, for at least 10 h. In each channel, the distance between the channel entrance and the tip of the longest growing hypha was measured. Channels presenting obvious signs of drying or of PDMS peeling off from the glass slide were excluded from analysis, as well as those already filled with a hypha longer than half the length of the channel at the beginning of the observations. Time-averaged elongation velocities $v_m$ were calculated as

$$v_m = \frac{x(t_f) - x(t_i)}{t_f - t_i},$$

where $t_i$ and $t_f$ denote the times at which the observation of the hypha started and stopped, and $x(t)$ represents the position of the hyphal tip at the time $t$. The maximum observation time $t_f - t_i$ is 10 h. For hyphae reaching the extremity of the channel earlier than 10 h, the last picture with a visible hyphal tip was taken into account. Spatio-temporal diagrams were generated using the built-in KymoGraph plugin to visualize elongation velocity variations. Image sequences of the two-dimensional growth of branched hyphae were manually thresholded, then analysed using the plugin Fractal Analysis. The Box Counting method was used to obtain the fractal dimension of each image in the sequence.

## 2.6. Statistical analysis

All results are presented as mean ± standard deviation. Whenever relevant, an unpaired Student's *t*-test was performed to compare results obtained in different experimental conditions. The confidence interval (99% or 95%) is indicated with the *p*-value in the legend.

# 3. Results

## 3.1. Fluid and mass transfer in absence of fungi

The chip designed for fungal culture was tested without mycelium and perfused with a 50 mg l⁻¹ fluorescein solution at a flow rate $Q = 9$ µl min⁻¹. Fluorescence intensity in both inoculation and perfusion chambers was monitored over time, close to the microchannels entrance, through time-lapse microscopy. In the perfusion chamber, the fluorescence intensity reached its maximal value, corresponding to a concentration of 50 mg l⁻¹, after 10 min of perfusion. In the inoculation chamber, the concentration remained well below this level, even after 150 min of perfusion. Figure 2*a* reveals that the concentration in the inoculation chamber started to increase fast after only 10 min. This characteristic time is shorter than the characteristic time $\tau_D$ for the diffusive transport of fluorescein $(D = 6.4 \times 10^{-8} \, \mathrm{m^2 \, s^{-1}})$ over the length $l = 1$ mm of the channels joining the chambers:

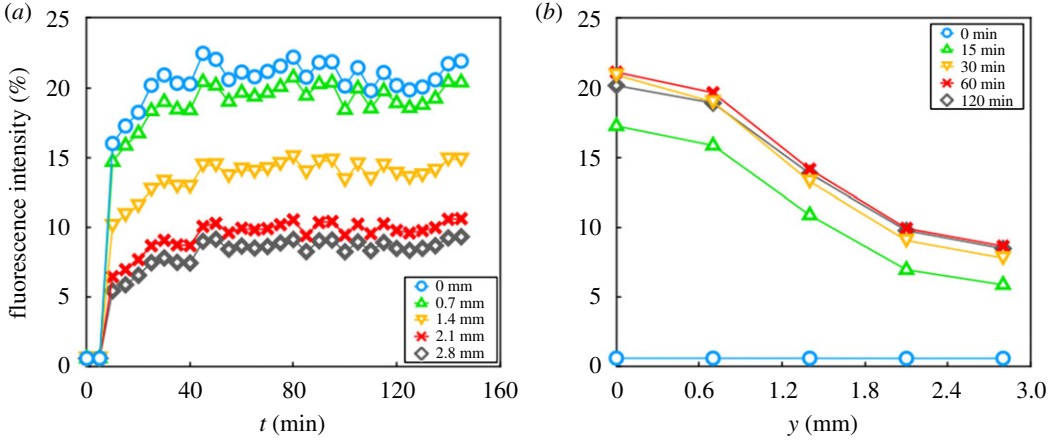

**Figure 2.** Measurements of the fluorescence intensity in the inoculation chamber, in the vicinity of the microchannels opening. (*a*) Fluorescence intensity over time for different distances *y* from the top of the chamber (see figure 1*a*). (*b*) Axial dependency of fluorescence intensity at different times.

$\tau_D = l^2/D = 26$ min, which is consistent with the fact that diffusion is not the sole mechanism transporting fluorescein across the microchannels and that convection accelerates the process. The fluorescence intensity in the inoculation chamber reached a plateau after 30 min for the next 2 h, and the plateau value depended on the distance *y* to the system entrance. The fluorescence intensity decreased with *y* in a manner that became time-independent after half an hour of perfusion, as can be seen in figure 2*b*. This variation along the *y*-axis can be explained by convective transport of the fluorescein solution across the microchannel. The medium flowing through the perfusion chamber creates a pressure drop between its inlet in $y = 0$ and its outlet in $y = L$, while the pressure in the inoculation chamber can be considered uniform. Close to the chip entrance, the pressure $P_p$ in the perfusion chamber is larger than the pressure $P_i$ in the inoculation chamber, and the fluid therefore flows from right to left through the microchannels between the two chambers. Conversely, when $y \leq L/2$, then $P_i \leq P_p$. In the microchannels of the bottom half of the chip, the fluid flows along the *x*-axis, from the inoculation towards the perfusion chamber. More details about the flow pattern in the chip can be found in the appendix.

## 3.2. Time-averaged elongation velocity of fungi in microchannels

For both *T. helicus* and *N. crassa*, the mycelium grew until it filled the inoculation chamber and reached the entrance of the microchannels. This initial phase took 3 to 4 days for *T. helicus*, and about 12 h for *N. crassa*. The diameter of *N. crassa* hyphae is about 10 µm, which is larger than the channel depth 5.8 µm, in contrast to *T. helicus* hyphae which are 2 to 5 µm wide depending on their age. Hyphae then started to extend into the microchannels and grew from the inoculation chamber to the perfusion chamber. A detail of *T. helicus* and *N. crassa* hyphae growing through a microchannel is pictured in figure 1*b,c*. We calculated the time-averaged elongation velocity $v_m$ for each individual hypha. All hyphae did not grow at the same speed, and figure 3 represents the distribution of elongation velocities for *T. helicus* (*a*) and *N. crassa* (*b*), in both static and dynamic conditions. While time-averaged velocities appeared to be consistent within a single chip, they varied significantly from one chip to the other, as indicated by electronic supplementary material, figure B2. This can be explained by inconsistent inoculum size, amount of nutrient-rich agar added with the inoculum and metabolic state: from one chip to the next, the amount of biomass and of nutrients inoculated in the chip was slightly variable. In spite of this variability, a clear difference in elongation velocities was found between the two strains. Without perfusion, $v_m$ was $190 \pm 46$ µm h$^{-1}$ on average for *N. crassa* and $47 \pm 19$ µm h$^{-1}$ for *T. helicus*.

For experiments in dynamic conditions, a flow rate of 1.8 µl min$^{-1}$, lower than in the previous section, was chosen in order to minimize shear stress on hyphae and medium consumption: with chamber dimensions of $w = 3$ mm by $h = 124$ µm, the average velocity $v_f$ of the fluid in the perfusion chamber was about 80 µm s$^{-1}$. The associated Reynolds number is defined as $Re = hv_f/v$, with $v = 10^{-6}$ m$^2$ s$^{-1}$ as the kinematic viscosity of water. Its value is of the order of $10^{-2}$. The maximal velocity in the microchannels at the beginning of the experiments is 11 µm s$^{-1}$, which is larger than the velocity of growing hyphae. Convective flows through the microchannels could possibly slow down hyphae

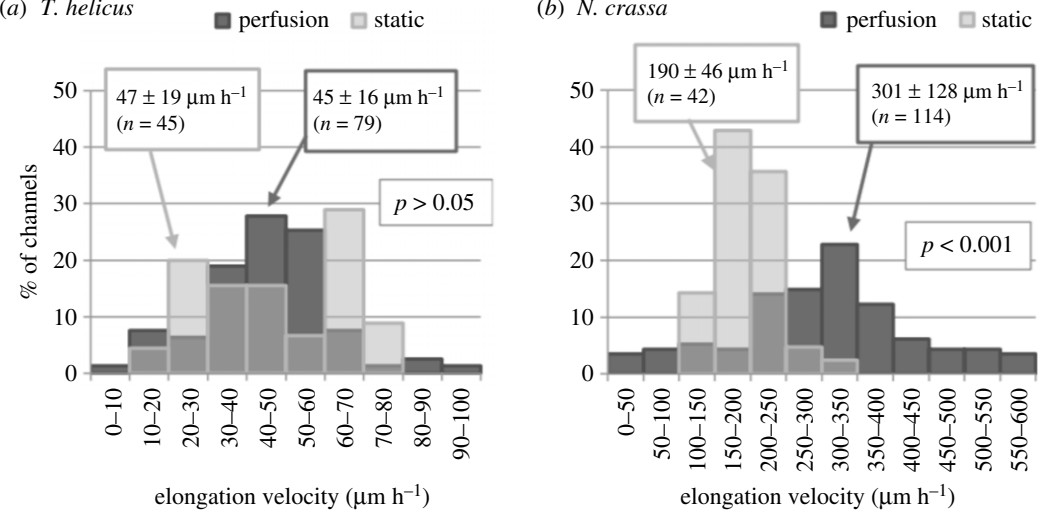

**Figure 3.** Distribution of hyphal elongation rates in microchannels, with and without perfusion for (a) *T. helicus* and (b) *N. crassa*. The mean elongation velocity ± standard deviation and the sample size are displayed in labels. Student's *t*-test was performed for each strain to compare static and dynamic conditions and the *p*-value is labelled on each graph.

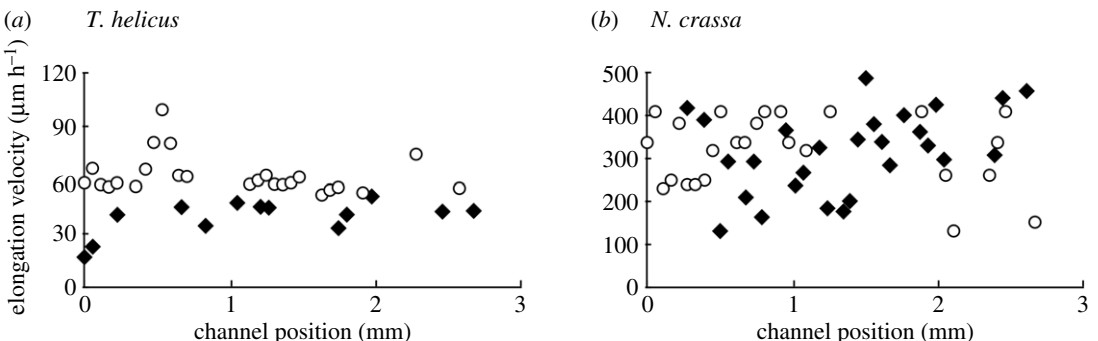

**Figure 4.** Hyphal elongation velocity measured in microchannels as a function of the distance *y* from the entrance of the perfusion chamber for (a) *T. helicus* and (b) *N. crassa*. For each strain, plain and hollow symbols represent two independent experiments.

growing in the opposite direction, close to the system entrance, or on the contrary accelerate hyphae growing in the microchannels closer to the system outlet, which could explain the variability of $v_m$. However, there was no noticeable influence of the channel position along *y* on the hyphal growth velocity $v_m$, as illustrated by figure 4.

As the distribution of elongation velocities displayed in figure 3(a) shows, there was no significant difference for *T. helicus* between chips incubated in static conditions and those with a circulation of medium ($v_m = 47 \pm 19\ \mu\text{m h}^{-1}$ and $45 \pm 16\ \mu\text{m h}^{-1}$, respectively). In dynamic conditions, mycelial growth in *N. crassa* was again faster than in *T. helicus*. Also, in contrast to *T. helicus*, growth velocities were significantly different for *N. crassa* between static and dynamic conditions: hyphae grew at $190 \pm 46\ \mu\text{m h}^{-1}$ on average without circulation of medium and $300 \pm 128\ \mu\text{m h}^{-1}$ with circulation.

## 3.3. Temporal variations of the elongation velocity

Time-lapse images acquired during confined growth were used to build spatio-temporal diagrams such as the one displayed in figure 5a. This graphical representation allows for rapid visualization of variations in elongation velocity. On such a diagram, growth at a constant rate appears as a straight line cutting diagonally from the top left to the bottom right of the figure, acceleration as a convex curve and deceleration as a concave curve. Hyphae sometimes stopped for a while and then resumed their growth. Pauses in *T. helicus* growth were usually associated with branching events, while in *N. crassa*, they were mostly observed when the cross-section of a microchannel was locally reduced due to a microfabrication defect. No further investigation of the temporal variations of *N. crassa* growth was

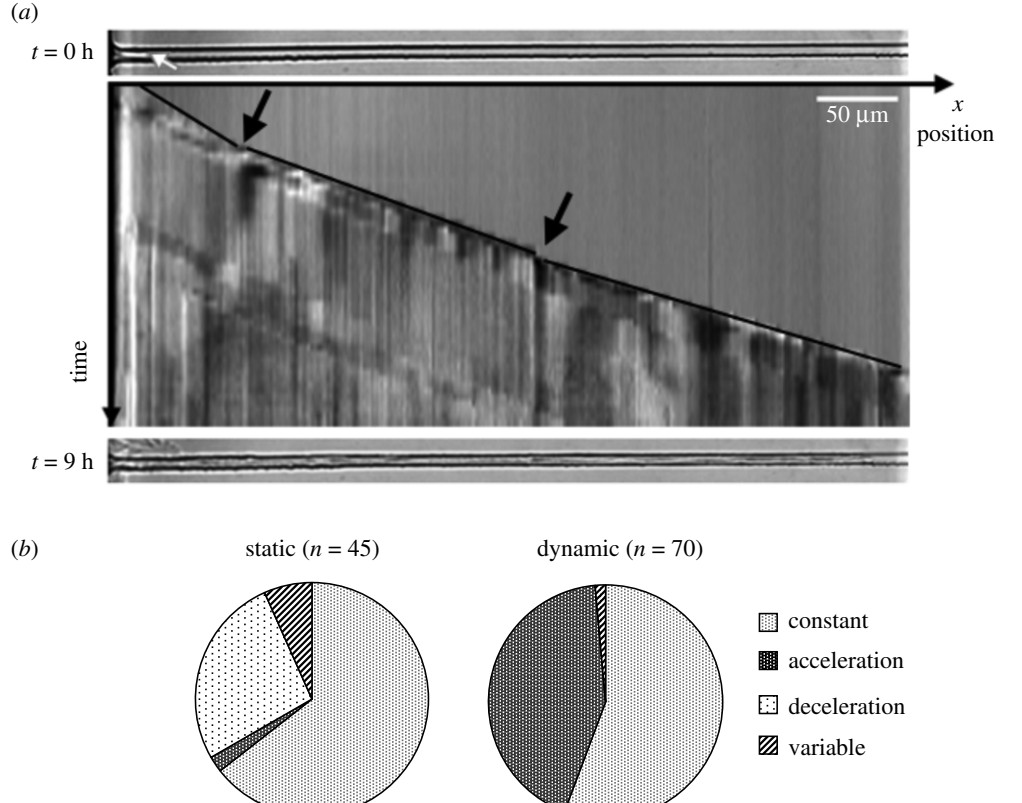

**Figure 5.** (a) Detail of the hyphal elongation of *T. helicus* in one microchannel over 9 hours, growing from left to right. Spatio-temporal diagram of hyphal length over time for a case of accelerating hyphal growth. Black lines were drawn to outline linear elongation phases. Slope changes and a brief elongation stop are indicated by black arrows. Micrographs of the empty channel in the initial state and of the channel filled with hyphae at the final state are shown on top of the diagram and at the bottom, respectively. (b) Distribution of the hyphal elongation profiles of *T. helicus* in microchannels in both static and dynamic conditions. In some channels, hyphae displayed successive acceleration and deceleration phases and were thus classified as 'variable' profiles.

performed, and we focused on *T. helicus*, whose thinner hyphae are much less sensitive to the irregularities in channel shape. In the example shown in figure 5a, *T. helicus* first grew at a constant rate, then accelerated. Accelerating hyphae are scattered along the system and not clustered close to (or far from) its entrance, and their average velocity is the same as the rest of the population. Figure 5b reveals that growth of *T. helicus* in static conditions tended to be mostly constant or decelerating, while velocity profiles noted in chips with perfusion were either constant or accelerating. The presence of a flow in the perfusion chamber thus appeared to influence the growth of *T. helicus*, although no difference between static and dynamic conditions was noticeable in the distributions of elongation velocity.

## 3.4. Petri dish growth

*N. crassa* and *T. helicus* were inoculated at the centre of Petri dishes and left to grow until they reached the edge of the dish. For both strains, the colony sizes were consistent between replicates and followed a similar trend: a short lag phase followed by a linear growth phase (figure 6a). The slope of these curves yields the radial growth velocity of the colony $v_c$, for which an almost 10-fold difference was found between the two strains. The average diameter of *T. helicus* colonies after 7 days was $33.4 \pm 0.1$ mm, which is consistent with values found in the literature (25–33 mm in 7 days) [16]. The colony growth velocity $v_c$ of 1040 µm h$^{-1}$ for *N. crassa* is consistent with values from the literature [17,18], and close to the value of 1440 µm h$^{-1}$ reported for unconstrained hyphal elongation rate of *N. crassa* [19]. Growth measured in colony experiments was always faster than in microfluidic experiments, as indicated by figure 6b, but the velocity ratio between micro- and macro-scale $v_m/v_c$ is smaller for *N. crassa*. Error bars represent the standard deviation, which is quite large due to the

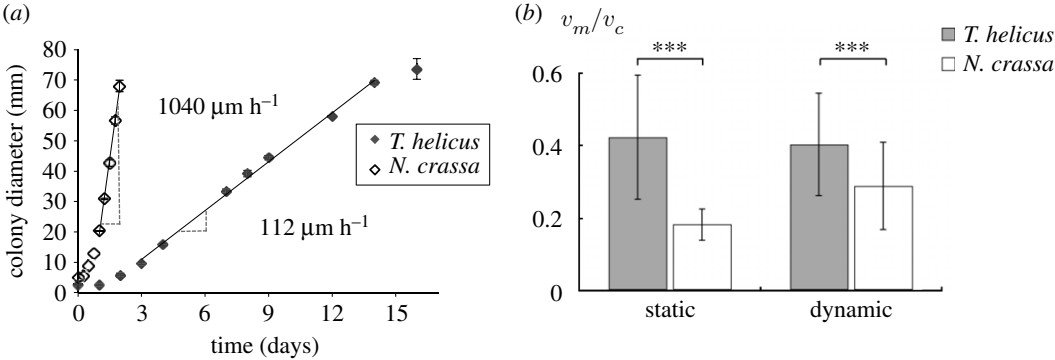

**Figure 6.** (*a*) Colony diameter over time for *N. crassa* and *T. helicus* grown on MYEA agar (average and standard deviation for triplicates). Labels indicate the colony radial growth velocities $v_c$ computed from the slopes of the linear fits. (*b*) Ratio between $v_m$, the velocity measured at microscopic scale in confined conditions, and $v_c$. In both static and dynamic conditions, the difference between *N. crassa* and *T. helicus* was significant ($p < 0.001$).

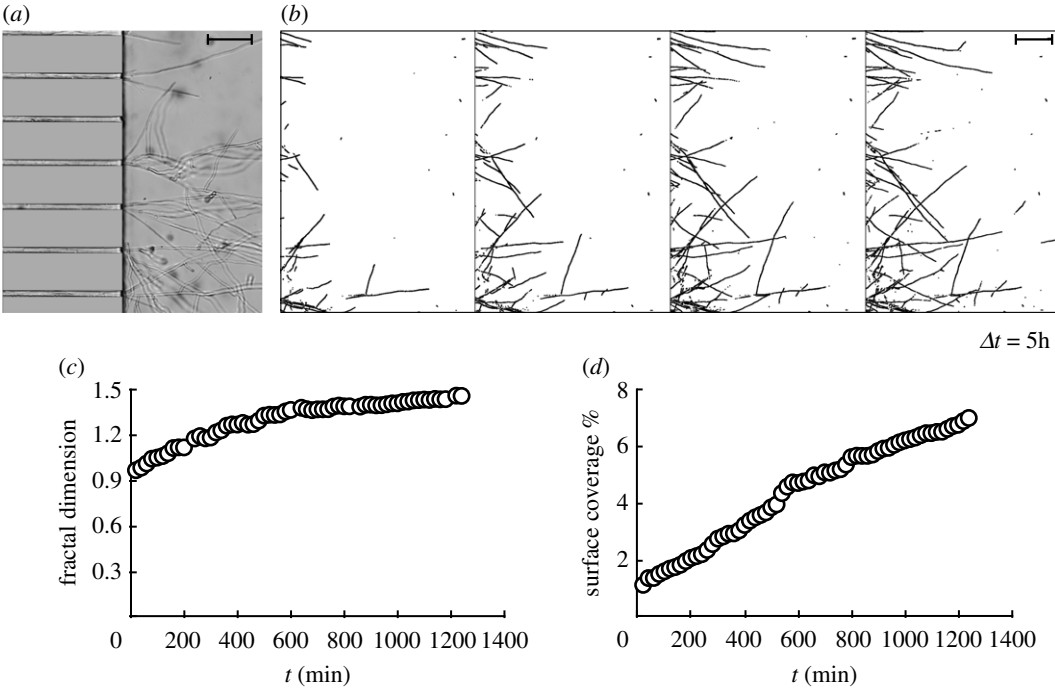

**Figure 7.** (*a*) Micrograph showing *T. helicus* hyphae growing out of the straight channel. Branching is visible. Scale bar, 100 μm (*b*) Time series showing the progression of two-dimensional growth in the chamber. Image threshold is adjusted to automatize measurements. (*c*) Fractal dimension as a function of time. (*d*) Surface coverage as a function of time.

number of measurements. Although error bars overlap in static conditions, Student's *t*-test indicates a significant difference between the two strains for both static and dynamic conditions.

## 3.5. Growth in two-dimensional chambers

As they reach the perfusion chamber, hyphae start to deviate from the microchannel axis, as illustrated in figure 7*a*. The tip velocity of single hyphae is not sufficient to predict the capacity of a fungal strain to colonize a soil, as a given total hyphal length can be associated with a diversity of mycelial morphologies [17]. Fractal geometry has been used for decades to described fungal growth. The fractal dimension *FD* characterizing mycelial growth on a surface is a number between 1 and 2. Automated image analysis can be performed as long as the mycelium is not too dense, either in a dilute suspension of mycelial fragments [20] at the early stages of pellet formation [21] or at the boundary of growing colonies [22]. Figure 7*b* shows how pictures obtained by time-lapse microscopy

were post-treated to extract the mycelial outline. Figure 7*c* presents the time evolution of fractal dimension. As hyphae started to grow out of the microchannels, the fractal dimension was close to 1, characteristic of linear growth, and increased over 10 hours to reach a value between 1.4 and 1.5. The surface occupied by the mycelium increased over time, as indicated by figure 7*d*.

# 4. Discussion

## 4.1. Water and nutrient transport

Geometries similar to the one presented here, featuring a row of microchannels joining two perfusion chambers, have been used to implement concentration gradients of soluble molecules with co-current or countercurrent perfusions. Using specific nutrients or signal molecules, such concentration gradients can be used to orientate cell growth or migration [23,24]. In experiments without fungi, we demonstrated that convection occurs between the perfusion and the inoculation chamber. The concentration gradient observed in figure 2*b* results from these convective flows in the microchannels. We showed that convection could accelerate fluorescein transport, in comparison with pure diffusive transport. In experiments involving fungi, however, the presence of hyphae in the microchannels modifies their hydraulic resistance. When all the microchannels contain hyphae, their permeability is expected to decrease dramatically. As microchannels become more resistive, convective flows between the two chambers slow down, or even stop, and no longer fuel the concentration gradient in the inoculation chamber. This means that when microchannels are occupied by hyphae, the main effect of the flow is to replenish the perfusion chamber with fresh medium, which feeds fungal growth mostly by diffusion. If the concentration of nutrients in the medium is large enough, then the growth conditions encountered by hyphae in all microchannels are comparable. This is consistent with the fact that the hyphal elongation velocity does not depend on the *y*-position of the channel (figure 4), even for *N. crassa*, which was shown to be sensitive to nutrient depletion. If the glucose flux was dependent on *y*, we would expect variations in the hyphal elongation velocity of *N. crassa*. This illustrates the fact that fungi have the ability to dramatically alter the permeability of a soil and influence water and nutrient transport.

Hyphal growth is associated with an intracellular water flux from older parts of the mycelium towards the growing apex [25,26]. Water flows have been characterized in *N. crassa* and shown to involve osmotic pressure gradients [27]. In our system, the inoculation chamber is a dead end. If the water that feeds growing hyphae originated from this compartment, a water flow in the opposite direction would be necessary. This water transfer could be extracellular and take place in empty channels, if there are any left, or in the occupied channels, in the interstitial space between the hyphae and the channel walls. Else, hyphae that were the first to cross the microchannels and stopped their growth could tap water from the perfusion chamber and be home to intracellular reverse water flow. In some of our experiments, all the microchannels were occupied by the mycelium and new hyphae kept growing between the two chambers. The presence of mycelium-free channels is thus not necessary. Whatever mechanism they use, fungi manage to grow from a water-limited to a water-rich compartment. Time-lapse observations did not provide any evidence of vesicle transport in the reverse direction, but only observations with an increased time resolution could totally rule out the intracellular water flow hypothesis, as the average value reported for protoplasmic flows in *N. crassa*, $2\,\mu\mathrm{m\,s^{-1}}$, is too fast to be observed with the time interval that was used in our experiments.

Apart from water, fungal growth requires nutrients such as glucose, ammonium, aminoacids, phosphates and trace elements, and gases such as oxygen, carbon dioxide and ammonia [28–30]. In response to the fluctuations of nutrient availability, fungi have developed strategies, such as the glucose transport system in *N. crassa* [31]. The different behaviour of *N. crassa* and *T. helicus* in on-chip experimental conditions, illustrated in figure 6*b*, could be explained by a difference in nutrient access. As previously mentioned, *N. crassa* and *T. helicus* display different hyphal diameters. This means that the degree of confinement, defined as the ratio between channel width and hyphal diameter, differs for both species. Figure 1*b,c* illustrates the morphological differences between *N. crassa* and *T. helicus*. Indeed, multiple hyphae of *T. helicus* can enter a microchannel simultaneously, while a single hypha of *N. crassa* occupies the whole channel width. Although the growth speed of *N. crassa* was larger than that of *T. helicus* in all the experiments, confinement had more impact on *N. crassa*, as observed in figure 6*b*: the ratio $v_m/v_c$ between confined hyphal growth rate and colony growth rate is larger for *T. helicus* than for *N. crassa*. As discussed earlier, convective nutrient transport is likely to be negligible

in our microfluidic fungal growth experiments, especially in the case of N. crassa whose hyphae almost entirely obstruct the microchannel cross-section, causing a dramatic increase in hydraulic resistance. Once a fungus has consumed the small amount of nutrients available in a microchannel, it has to rely on diffusion only or on its own intracellular reserves to sustain its growth. But since the diffusion coefficient of glucose ($0.6 \times 10^{-11}\,\mathrm{m^2\,s^{-1}}$) [32] is $100\times$ smaller than that of fluorescein ($0.64 \times 10^{-9}\,\mathrm{m^2\,s^{-1}}$) [33], diffusion is $100\times$ slower for glucose than for fluorescein, yielding a characteristic time for glucose diffusion over the channel length of about 10 h, which is long in comparison with typical times for N. crassa growth. The growth of N. crassa over a given length requires more energy than that of T. helicus: because its hyphae are thicker, the amount of biomass corresponding to this length is larger. Also, N. crassa grows faster than T. helicus, which is another reason why its nutrient consumption rate is expected to be larger than that of T. helicus. It thus seems that the growth of N. crassa is limited by the flux of nutrients diffusing from the perfusion chamber in static and probably also in dynamic conditions. In contrast, nutrient availability does not appear to significantly affect the growth of T. helicus in these conditions, since enhancing nutrient supply through perfusion does not modify its elongation velocity. Lack of space due to the geometric constraints and limited access to oxygen may thus be the main limiting factors for this last strain. The relatively slow growth of T. helicus seems to be associated with a moderate consumption of nutrients and its needs are met with the initial amount of nutrients in the chamber.

## 4.2. Effect of confinement

Differences in nutrient consumption rate between the two strains might explain why N. crassa is more affected than T. helicus by nutrient deprivation in microfluidic conditions, but it does not explain why the elongation velocity of both strains is so small in comparison with the growth velocity measured at the scale of the colony. It should be noted that MYEA, which is used for macroscopic scale experiments, is a complex medium as opposed to the MMG medium infused in the microsystems and thus provides nutrients in a different chemical form. However, in the case of T. helicus, elongation velocity is not affected by nutrient supplementation through a perfusion, which suggests that the availability of essential nutrients is not the main factor limiting its growth.

Spatial limitations in the microfluidic chips may also explain this slower growth. It could be due to the detection of the microchannel surfaces by hyphal membranes, since several types of mechanosensitive receptors control fungal morphogenesis including hyphal growth [34]. Notably, for N. crassa, thigmotropism has been proposed to be due to a putative calcium channel protein responding to mechanical stresses at the apex and triggering changes in the polarity machinery at the hyphal tip [35].

The presence of a hypha in a channel did not automatically prevent new hyphae coming from the inoculation chamber to colonize the channel. Simultaneous growth of multiple hyphae happened more frequently with T. helicus than with N. crassa. The second hypha generally grew more slowly than the first one, but the amount by which it was slowed down differed depending on the strain: the velocity decreased by 16% in T. helicus and 69% in N. crassa. This again illustrates the fact that N. crassa, being more confined in the channel than T. helicus, is also more sensitive to an additional confinement.

For T. helicus, branching events were observed in the straight channels and usually associated with a brief pause or a decrease in elongation rate. This arrest of mycelial growth prior to branching events was also observed in the unconfined growth of single hyphae [36]. We did not observe branching of N. crassa during its growth in the microchannels. Measurements of the branching distance of N. crassa in tortuous microsystems are found in the literature: it decreases from 219 μm on an agar plate to 43 μm in a microfluidic network of $12 \times 10$ μm channels and even to 23 μm in complex maze-like structures [18]. This small branching distance was interpreted as a strategy for the fungus to probe its environment and solve the maze when faced with obstacles. Our observations suggest that N. crassa is also capable of adjusting its branching distance to the straight geometry of our channels, suppressing branching over lengths exceeding its unconfined branching distance. In soil, this could allow the fungus to rapidly follow cracks in rocks. To the best of our knowledge, no values of the fractal dimension characterizing the growth of T. helicus are available in the literature, but our experimental values are consistent with those reported for other strains.

The quasi-linear shape of the curve in figure 7d is remarkable. Indeed, if all the hyphae in the branched network were to grow with the same velocity as the initially straight hyphae, surface coverage should increase in a faster, nonlinear manner. But we observe that, as the network density increases, some hyphae stop completely while others keep growing. This is likely to cause, on average, a decrease in the apical velocity. Note that the thresholding method used to analyse our images does not allow us to

distinguish between intersecting and bifurcating hyphae, which prevents us from tracking the elongation velocity of individual hyphae over time, but we can still evaluate the instantaneous tip velocity of several apices in the same field of view. This decrease in average apical velocity seems to almost equally compensate the increase in the number of hyphae. This arrested growth could be seen as an illustration of a quorum sensing phenomenon between hyphae, which is poorly understood and studied in fungi, but has been shown to exist in *N. crassa* and to control fungal growth in other species [37]. More precisely, quorum sensing limits fungal growth in order to regulate cell density and avoid competition for nutrients [30]. A seemingly related behaviour was also observed in some microchannels where fungal growth started late: the simultaneous growth of two hyphae in opposite directions, one of them originating from the perfusion chamber that had been reached by faster neighbours. In this case, a pause was observed long before the two hyphae were in contact, then one of them resumed it growth towards the other one. In macroscopic experiments in Petri dishes, because there is no vertical confinement, hyphae can avoid each other without needing to stop.

# 5. Conclusion

The transport of solutes through microchannels was characterized in the microfluidic device in the absence of fungus, showing that stable concentrations in the culture chambers can be achieved for several hours. The culture of two filamentous fungal strains was achieved in the microfluidic chambers, in presence or absence of a perfusion, and time-averaged elongation velocities were measured. The size of the channels connecting the two chambers is comparable to that of small inter-aggregate spaces or large intra-aggregate pores, which is a relevant scale, in soils, for both water transport and soil–microbe interactions [5]. Our system was well adapted to create a stable environment for the slow-growing strain *T. helicus*, while the model strain *N. crassa* was limited in its growth by the low nutrient supply in confined conditions. This suggests that *N. crassa*, although widely used in the literature due to its being easy to handle in the lab, is more sensitive to confinement. Experimental results obtained in unconfined conditions should therefore be handled with care, as they may not always be transferable to predict fungal behaviour in soil. The distance of each channel from the perfusion inlet had no effect on elongation velocities, suggesting that all channels provide equivalent conditions for hyphal growth. This also suggests that the presence of fungi in a soil significantly alters its permeability and thus water and nutrient transport. Given the high variability in elongation velocities for the same strain in the same conditions, this device proved useful to study the behaviour of individual hyphae with a high number of replicates. For both strains considered, hyphal elongation in microfluidic channels was always slower than unconstrained radial colony growth on agar. We showed that this reduced velocity can be explained in part by the limited amount of nutrients available in microsystems, but that confinement also has an effect on fungal behaviour. When growing in microchannels, the branching distance of *N. crassa* increased beyond its unconfined value. Qualitative observations of mutual interactions between hyphae in confined one- and two-dimensional geometries suggest that hyphal growth inhibition may also contribute to the reduction in growth velocity observed at the microscale. In the future, we hope that a more systematic study of growth velocities across scales will allow to extract experimental values for the parameters involved in models of fungal growth in silico, in order to better predict the behaviour of fungi in soils [38].

Ethics. Investigations were carried out in full accordance with the ethical guidelines of our research institution.
Data accessibility. Our data are accessible at the Dryad Digital Repository: https://doi.org/10.5061/dryad.76hdr7ss1 [39].
Authors' contributions. C.B., A.F. and A.L.G. designed the research. C.B. performed the research. C.B., A.F. and A.L.G. analysed the data and wrote the paper. All authors gave final approval for publication.
Competing interests. The authors have no competing interests.
Funding. C.B. benefited from a doctoral grant from the French Ministry of Research. The MycoFlu project has been supported by grants from the CNRS (EC2CO programme MICROBIEN) and Sorbonne Universités (programme Emergence).
Acknowledgements. The authors would like to thank Xue Sun, Alicia Alejandra Mier Gonzalez, Marie Valmori, Roxane Valentin and Théo Guillerm for their help with the preliminary experiments.

# Appendix A. Fluid velocity in the parallel microchannels

The mean velocity $\bar{v}$ in a rectangular channel is defined by the flow rate $q$ divided by the cross-section $S$,

$$\bar{v} = \frac{q}{S}. \tag{A 1}$$

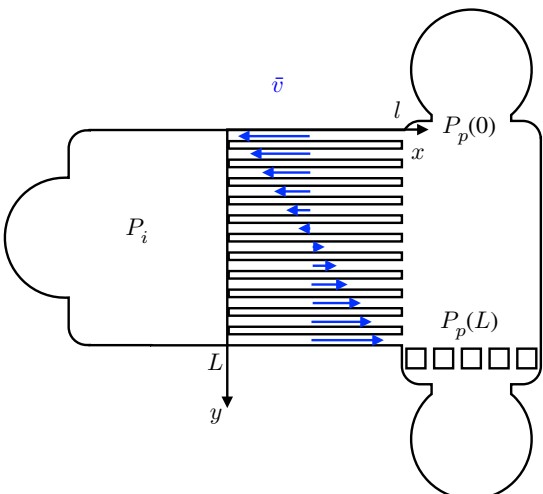

**Figure 8.** Sketch of the device. The blue arrows indicate the direction and amplitude of the fluid velocity in the parallel microchannels.

The flow rate $q$ is proportional to the pressure difference $\Delta P$ between the entrance and the exit of the channel, as Poiseuille discovered for cylindrical pipes, and the prefactor is called hydraulic resistance, $\Delta P = R_h q$. The hydraulic resistance of a shallow rectangular microchannel can be expressed as a function of its length $l$, its width $w$, its depth $h < w$ and the viscosity $\mu$ of the fluid [40],

$$R_h = \frac{12\mu l}{h^3 w}\left[1 - \Sigma_{n,\text{odd}}\frac{192}{(n\pi)^5}\frac{h}{w}\tanh\left(n\pi\frac{w}{2h}\right)\right]^{-1}. \tag{A 2}$$

Let us denote by $P_p$ and $P_i$ the pressures in the perfusion and inoculation chambers, respectively. If we assume that the leaks through microchannels represent only a small perturbation of the main flow in the perfusion chamber, we expect $P_p$ to be uniform in a cross-section of the chamber and to decrease with $y$ in a linear manner,

$$P_p(y) = P_p(y = 0) - \frac{y}{L}\Delta P_p, \tag{A 3}$$

with $\Delta P_p = P_p(y = 0) - P_p(y = L)$ the pressure difference between the inlet and the outlet of the perfusion chamber, computed using equation (A 2) with the dimensions $W \times H \times L$ of the perfusion chamber and the flow rate $Q$ imposed by the pump. Under the same hypothesis, in the inoculation chamber, the fluid is almost at rest and the pressure $P_i$ can be considered constant.

All the microchannels have the same dimensions $w \times h \times l$ and thus the same hydraulic resistance $R_{h,\text{channel}}$. Each of them is located at a different distance $y$ from the entrance of the flow chamber (figure 8) and exposed to a pressure difference $P_p(y) - P_i = P_p(0) - P_i - (y/L)\Delta P_p$, from which we can compute the flow rate through the microchannel towards the inoculation chamber,

$$q(y) = \frac{P_p(y) - P_i}{R_{h,\text{channel}}} = \frac{P_p(0) - P_i}{R_{h,\text{channel}}} - \frac{\Delta P_p}{R_{h,\text{channel}}}\frac{y}{L}, \tag{A 4}$$

$q$ decreases linearly with $y$. Knowing that the inoculation chamber has no inlet or outlet apart from the microchannels, the amount of fluid flowing in and out of the microchannels has to be null. This implies that $q(y = L/2) = 0$ and allows to compute $P_i = P_p(0) - \Delta P_p/2$. The flow rate in the microchannel reaches its maximum when $y = 0$ and $y = L$, close to the inlet and outlet of the perfusion chamber,

$$q_{\max} = \frac{\Delta P_p}{2R_{h,\text{channel}}} = Q\frac{R_{h,\text{chamber}}}{2R_{h,\text{channel}}}. \tag{A 5}$$

Finally, the mean velocity in the microchannel, which is proportional to $q$, is equal to 0 at the middle of the chamber and maximal in $y = 0$ and $y = L$, as sketched in figure 8.

**Table 1.** Dimensions and characteristics of flow in the microchannels and in the perfusion chamber.

|  | microchannels | perfusion chamber |
|---|---|---|
| $w$ (μm) | 10 | 3000 |
| $h$ (μm) | 6 | 124 |
| $l$ (μm) | 1000 | 3000 |
| $R_h$ (S.I.) | $8.88 \times 10^{15}$ | $6.46 \times 10^9$ |
| $q$ (μl min$^{-1}$) | $6.55 \times 10^{-7}$ | 9 |
| $\bar{v}$ (mm min$^{-1}$) | 0.05 | 24.19 |

We can compute its maximal value,

$$\bar{v}_{\max} = \frac{Q}{hw} \frac{R_{h,chamber}}{2R_{h,channel}}. \tag{A 6}$$

In table 1, we present the experimental parameters used to compute the numerical values of $\bar{v}_{\max}$. We check that the flow rate $q$ in the microchannels, even multiplied by the number $N = 50$ of parallel channels, is indeed negligible compared with the perfusion rate $Q$. Given the dimensions of microchannels and the range of flow rates, the assumptions made at the beginning of the calculation (unperturbed perfusion flow, negligible pressure drop in the inoculation chamber) are thus valid. We can conclude that the fluid is almost at rest in the inoculation chamber. As long as the microchannels remain devoid of hyphae, the velocity depends of $y$ in a linear manner. This velocity is much smaller than the average velocity in the perfusion chamber. At such a flow rate, removing all the fluorescein-poor fluid from the inoculation chamber would take between 9 and 10 days. As the fungus develops, hyphae obstruct the microchannels that become even more resistive, further decreasing the rate of fluid transport between the chambers. This indicates that the proposed geometry can be used to implement concentration gradients across the microchannels, that can be maintained over several days.

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
