## [Reviewer comments · Royal Society Open Science]

Review History

RSOS-191535.R0 (Original submission)

Review form: Reviewer 1

Is the manuscript scientifically sound in its present form?

No

Are the interpretations and conclusions justified by the results?

No

Is the language acceptable?

No

Do you have any ethical concerns with this paper?

No

Have you any concerns about statistical analyses in this paper?

No

Recommendation?

Major revision is needed (please make suggestions in comments)

Comments to the Author(s)

The manuscript deals with the growth of two filamentous fungi in a microfluidic device: *Talaromyces helicus* (which can be used in bioremediation of hydrocarbons and heavy metals) and *Neurospora crassa* (a “model” filamentous fungus). Unfortunately, it is not very clear how the manuscript advances scientific knowledge. Do the advances have something to do with the design of the microfluidic device? If so, the advantages of the device were not properly discussed in the context of the literature regarding the growth of filamentous fungi in microfluidic devices. Do the advances relate to the particular results that were obtained with the two fungi? If so, is not clear whether the comparison between *T. helicus* and *N. crassa* serves any useful purpose. Additionally, although the authors suggest that their microfluidic device can mimic the confinement that occurs within soils, it is not clear whether their results have any real implications for understanding fungal growth in soils. Part of the problem is that the discussion is quite speculative and imprecise. In many places, it is not very clear what the authors are trying to say. The discussion needs to be completely rewritten – the authors should limit themselves to conclusions that are supported by the data and also need to write in a clearer, more precise manner. In general, the paper needs to make it explicitly clear exactly what is the advancement of scientific knowledge that it contributes - in the context of the current literature in the area of that advancement.

Please note that I made some comments directly onto a PDF of the manuscript, which I uploaded onto the system (Appendix A). However, the authors need to realize that it is not simply a case of addressing these comments one by one - I could have made many, many more comments than I did.

Review form: Reviewer 2

Is the manuscript scientifically sound in its present form?

No

Are the interpretations and conclusions justified by the results?

No

Is the language acceptable?

Yes

Do you have any ethical concerns with this paper?

No

Have you any concerns about statistical analyses in this paper?

No

Recommendation?

Major revision is needed (please make suggestions in comments)

Comments to the Author(s)

Please see the attached file (Appendix B) for comments to the authors.

Decision letter (RSOS-191535.R0)

18-Oct-2019

Dear Dr Le Goff,

The editors assigned to your paper ("Microfluidic monitoring of the dynamic behavior of individual hyphae in confined environments") have now received comments from reviewers. We would like you to revise your paper in accordance with the referee and Associate Editor suggestions which can be found below (not including confidential reports to the Editor). Please note this decision does not guarantee eventual acceptance.

Please submit a copy of your revised paper before 10-Nov-2019. Please note that the revision deadline will expire at 00.00am on this date. If we do not hear from you within this time then it will be assumed that the paper has been withdrawn. In exceptional circumstances, extensions may be possible if agreed with the Editorial Office in advance. We do not allow multiple rounds of revision so we urge you to make every effort to fully address all of the comments at this stage. If deemed necessary by the Editors, your manuscript will be sent back to one or more of the original reviewers for assessment. If the original reviewers are not available, we may invite new reviewers.

- Data accessibility

<http://datadryad.org/submit?journalID=RSOS&manu=RSOS-191535>

- **Competing interests**

- **Authors' contributions**

- **Acknowledgements**

- **Funding statement**

Kind regards,
Anita Kristiansen
Editorial Coordinator
Royal Society Open Science
openscience@royalsociety.org

on behalf of Professor Guy Genin (Associate Editor) and Pietro Cicuta (Subject Editor)
openscience@royalsociety.org

Associate Editor's comments (Professor Guy Genin):

Comments to the Author:

The associate editor is enthusiastic about the study. However, as both reviewers note, the paper will need substantial rewriting to meet the requirements of this particular journal. The associate editor encourages the authors to consider making these substantial changes and resubmitting.

The key points raised by the reviewers are that:

- (1) all conclusion and discussion points need to be clearly supported by the results; for example, the claim that the system replicates soil needs to be supported or deleted
- (2) although novelty is not required, the results do need to be placed in the context of the state of the art, for example in the context of microfluidic growth systems and hyphae growth

(3) given the absence of a page limit for this journal, the authors might consider moving all results that are part of their main narrative to the main manuscript, and reserving the supplement only for results that are tangential

Reviewers' Comments to Author:

Reviewer: 1

Comments to the Author(s)

The manuscript deals with the growth of two filamentous fungi in a microfluidic device: *Talaromyces helicus* (which can be used in bioremediation of hydrocarbons and heavy metals) and *Neurospora crassa* (a "model" filamentous fungus). Unfortunately, it is not very clear how the manuscript advances scientific knowledge. Do the advances have something to do with the design of the microfluidic device? If so, the advantages of the device were not properly discussed in the context of the literature regarding the growth of filamentous fungi in microfluidic devices. Do the advances relate to the particular results that were obtained with the two fungi? If so, is not clear whether the comparison between *T. helicus* and *N. crassa* serves any useful purpose. Additionally, although the authors suggest that their microfluidic device can mimic the confinement that occurs within soils, it is not clear whether their results have any real implications for understanding fungal growth in soils. Part of the problem is that the discussion is quite speculative and imprecise. In many places, it is not very clear what the authors are trying to say. The discussion needs to be completely rewritten – the authors should limit themselves to conclusions that are supported by the data and also need to write in a clearer, more precise manner. In general, the paper needs to make it explicitly clear exactly what is the advancement of scientific knowledge that it contributes - in the context of the current literature in the area of that advancement.

Please note that I made some comments directly onto a PDF of the manuscript, which I uploaded onto the system. However, the authors need to realize that it is not simply a case of addressing these comments one by one - I could have made many, many more comments than I did.

Reviewer: 2

Comments to the Author(s)

Please see the attached file for comments to the authors.

Author's Response to Decision Letter for (RSOS-191535.R0)

See Appendix C.

RSOS-191535.R1 (Revision)

Review form: Reviewer 3

Is the manuscript scientifically sound in its present form?

No

Are the interpretations and conclusions justified by the results?

No

Is the language acceptable?

Yes

Do you have any ethical concerns with this paper?

No

Have you any concerns about statistical analyses in this paper?

No

Recommendation?

Major revision is needed (please make suggestions in comments)

Comments to the Author(s)

The present manuscript presents a study on the extension rate of individual hyphae in a microfluidic device. Although the authors have collected many data, there are two main problems with the manuscript.

Firstly, the objective of the work is not clear. The abstract seems to indicate that the purpose of the study was to determine if it is acceptable to use these types of devices to study growth of single hypha. However, throughout the manuscript, data are presented indicating that growth in the device is slower than growth on agar and the authors do not address this matter properly. In the previous revision, reviewers indicated that the novelty of the work was unclear and, nevertheless, the authors have not explained their objective in the introduction section. The authors replied "The design of the microfluidic system is indeed not new. The objective of this work was to study the growth of a large number of hyphae in confined conditions, with two different degrees of confinement, in order to better understand why growth velocity varies across scales." However, this information is not in the manuscript and is vague. What do the authors mean by two different degrees of confinement? What scales are they referring to?

Secondly, the authors conclude that there is nutrient depletion in the experiment and this explains why *N. crassa* is more affected by the microchannels, but the medium used in the experiments does not contain a carbon source, therefore, the nutrients used for growth are either inside the hyphae or in the inoculum agar, thus, it is in the inoculum chamber. Consequently, the transport of the mineral medium from the other chambers would unlikely affect growth. This is a main aspect of the discussion that needs to be reviewed. Many arguments made by the authors are not validated by the results.

A detailed list of suggestions is presented below:

- 1) Title: "Microfluidic monitoring of the dynamic behavior of individual hyphae in confined environments". What do you mean by dynamic behavior? Growth rate is mostly constant in the experiments.
- 2) Abstract: "The distributions of growth velocity obtained for each strain were compared to measurements obtained in less confined geometries, either in 2D chambers". Where in the manuscript is the comparison between growth in the channel and in the 2D chambers?
- 3) Figures: it is important that figures are easy to understand based only on the figure itself and its legend, without reading the text. In order to achieve this, I present some suggestions:
 - a) Figure 1: What does "10 x 6 μm " stand for? Are the microchannels cylindrical? Is it 6 or 5.8 μm ?
 - b) Figure 4: Legend should read "Hyphal elongation velocity measured in a microchannel AS a function of the distance y from the entrance of the perfusion chamber". Also, indicate the meaning of each symbol in the legend or in the figure.
 - c) Figure 5b: include a title in y-axis explaining what the percentage means. Also, the expression "Temporal evolution of *T. helices* hyphal growth" used in the legend is vague and does not indicate what data was used to construct the graph. It might help to indicate in the

figure how the profile in Figure 5a would be classified. It would be better to use the same terms used in the text in the x-axis: accelerating and decelerating.

- d) Figure 6a: replace the y-axis title for "colony diameter".
- e) Figure 6b: Include statistical analysis in (b). In figure legend, explain the symbols v_m and v_c .
- f) Figure 10: Indicate in the Figure legend what is A, B and C
- g) Check that all figures are cited in the text.
- h) Check the numbering of figures and tables that are not in the main text.

4) Introduction:

- a) Is there a reason why it is a single paragraph? It is usually separated into 3 to 5 paragraphs, with each paragraph with an idea. Consider fragmenting the text into more paragraphs.
- b) Line 25: "In particular, some concerning organic pollutants can be used as carbon and energy source by strains able to access to these molecules and degrade them." It should be "strains able to access and degrade these molecules"
- c) Line 39: remove the space before the comma "per hour ,"
- d) Line 49: If the device was developed by the authors and is presented for the first time in this paper, it would be better to write "In this study, we propose a microfluidic device to monitor the growth of individual hyphae in microchannels." If the device has been used before, there should be a citation for the previous work.
- e) Line 51: "This set-up aims at mimicking a conned, heterogenous and dynamic environment," The terms "heterogenous" and "dynamic" can mean a lot of different things in terms of devices for microfluid studies. I suggest you explain what kind of studies can be performed with this device.
- f) Line 52: Explain which are the "both strains" used in the work, because it is unclear at this point.
- g) What questions are you trying to answer with your research? At the end of the introduction, present the objective of the paper. In the present version, you only present the purpose of the device.

5) Experimental section:

- a) Does the *Talaromyces helices* strain have a strain code? Has this strain been used in previous published works?
- b) Line 73: "The intensity of fluoescence in the chambers was monitored by fluoescence microscopy over 3 h using the FITC filter". Where exactly was fluoescence measured? In the entire chamber? In the microchannels? In the outlet? Perhaps indicate in the figure of the chamber the points of analysis.
- c) Line 76: Inform the composition of the mineral medium or cite a reference to the composition.
- d) Section 2.5: What medium was used? What temperature?
- e) Section 2.5: "maximum and minimum diameter of the colony were measured using ImageJ". For each image, only 1 value of maximum diameter and 1 value of minimum diameter were taken? Why was the median value used instead of the average?
- f) Section 2.5: In "Time-averaged elongation velocities were calculated as", include the symbol v_m .
- g) All image analysis was performed in ImageJ? This is not clearly stated.

6) Results

- a) What are the hyphal diameters of the strains tested when grown without confinement? Figure 1b indicates that *N. crassa* is larger than the microchannels. How does it affect growth? These information are given in the discussion but should be presented here.
- b) The data presented in Figure 2 was measured in the inoculation chamber or the microchannels? What does the position y refer to? It would be better to indicate the y positions in the figure of the device.

- c) Line 103: The radius of the perfusion chamber is 1 mm, which is the length the authors used to calculate the characteristic time. The experimental time is shorter, which is expected if the measurement was done in a radius lower than 1 mm and if convective flow occurs in the chamber.
- d) Line 106: "Figure 2(a) reveals that the concentration in the inoculation chamber increased fast in the first 30 min, and reached a plateau for the next two hours.". According to Fig 2a, it actually increased fast in the first 10 min and, based on fig 2b, it reached a plateau in 30 min (the curves in fig 2b overlap after 30 min).
- e) Line 109: "This axial dependency can be explained by convective transport of the fluorescein solution through the microchannel." This sentence is misleading. The authors refer to axial dependency in the chamber caused by flow in the microchannel, but it seems that it refers to axial dependency in the microchannel. The authors need to improve the added text in lines 109 to 113, because it is only understandable after reading the appendix.
- f) Line 122: How does inoculum size affect the velocity of individual hyphae? The metabolic state hypothesis might be valid, in addition to compounds from the media diffusing to the perfusion, but the inoculum size does not affect individual hyphae.
- 7) Discussion:
- a) Line 178: "When all the microchannels contain hyphae, we expect their permeability, and as a consequence the gradient in the inoculation chamber, to vanish" Do you mean the fluid will not reach the inoculation chamber when hyphae are present in the channels? This may be true for *N. crassa*, which seems to occupy the entire channel width, but not for smaller hyphae.
- b) "The fact that hyphal elongation velocity does not depend on the channel position y , even when some microchannels remain empty, seems to suggest that this situation occurs even before all the channels are filled with hyphae." I don't think you can conclude this. Since there is no difference in extension rate between the static and the perfusion experiment, I believe the nutrients are not being depleted in the static or perfusion experiment. In fact, the medium used in the perfusion does not contain a carbon source, which is the nutrient whose lack of limits growth. The nutrients present in the mineral medium would unlikely be depleted because only small amounts of them are used. Therefore the only effect of the flow would be frictional or removal of biological compounds related to quorum sensing or the mineral medium is washing out the nutrients from the inoculum. In order for you to conclude that there is no flow in the chambers, you should have replaced the circulating buffer with fluorescein after the hyphae had occupied the channels. The nutrients in mineral media are not enough for growth.
- c) Line 200: "Variations in elongation velocities between experimental conditions could be explained by a modification of nutrient availability". Which experimental conditions? Which variations in elongation?
- d) Line 211: There is a discussion about glucose, but there is no glucose in the medium used. Hyphae are either using carbon from the agar of the inoculum or intracellular reserves.
- e) Line 217: it is also limited by the lack of space.
- f) "It thus seems that the growth of *N. crassa* is limited by the flux of nutrients diffusing from the perfusion chamber, in static and probably also in dynamic conditions". But there is no carbon source in the perfusion chamber.
- g) "Its metabolic rate may already be maximal given the geometric constraints." I don't understand this sentence.
- h) Line 224: "Differences in nutrient consumption rate between the two strains might explain why *N. crassa* is more affected than *T. helicus* by nutrient deprivation in microfluidic conditions". I disagree given what is written here. The probable explanation is lack of space, since the diameter of *N. crassa* is larger than the size of the channel.
- i) Line 225: "elongation velocity of both strains is so small in comparison with the growth velocity measured at the scale of the colony". The authors can't compare growth in agar with confined growth. In agar, there is substrate (carbon source) available under the colony and O₂, which are lacking in the device. In addition, there are walls blocking hyphal growth in the device.
- j) Figure 7 should be in the results section.
- k) You could check if it is the wall effect by measuring linear extension rate after the channels and comparing.

8) Conclusion:

a) Line 280: "This suggests that *N. crassa*, although widely used in the literature, may not be the best experimental model for studies related to soil." I don't understand how you can conclude this.

9) Appendix:

a) What does y represent? Is it position in the microchannels? Please cite the corresponding figure.

b) Figure 8 should be figure A1 because it is in the appendix. Table should be numbered A1.

c) Line 317: Pressure drop refers to a loss in fluid pressure between 2 points. In "pressure drop in the inoculation chamber", what are the two points? In "pressure drop in the perfusion chamber", what are the two points?

d) Line 318: write constant instead of cst.

e) Line 331 "This is in agreement with the assumptions made at the beginning of the calculation (unperturbed perfusion flow, negligible pressure drop in the inoculation chamber)." Ending the appendix with this statement gives the wrong idea that the assumptions used as starting point are only confirmed by the deduction made, which is not entirely true and which does not prove that the assumptions are true because they were input to the deduction. Whenever possible, assumptions should be proved in the middle of the deduction (e.g. that flow in the channels is insignificant compared to flow in the large chamber).

Review form: Reviewer 4

Is the manuscript scientifically sound in its present form?

Yes

Are the interpretations and conclusions justified by the results?

Yes

Is the language acceptable?

Yes

Do you have any ethical concerns with this paper?

No

Have you any concerns about statistical analyses in this paper?

Yes

Recommendation?

Accept as is

Comments to the Author(s)

I confirmed the author's revision is proper and recommend to be published in this journal.

Decision letter (RSOS-191535.R1)

26-Feb-2020

Dear Dr Le Goff:

Manuscript ID RSOS-191535.R1 entitled "Microfluidic monitoring of the dynamic behavior of individual hyphae in confined environments" which you submitted to Royal Society Open Science, has been reviewed. The comments of the reviewer(s) are included at the bottom of this letter.

Please submit a copy of your revised paper before 20-Mar-2020. Please note that the revision deadline will expire at 00.00am on this date. If we do not hear from you within this time then it will be assumed that the paper has been withdrawn. In exceptional circumstances, extensions may be possible if agreed with the Editorial Office in advance. We do not allow multiple rounds of revision so we urge you to make every effort to fully address all of the comments at this stage. If deemed necessary by the Editors, your manuscript will be sent back to one or more of the original reviewers for assessment. If the original reviewers are not available we may invite new reviewers.

- Ethics statement

- Data accessibility

- Competing interests

- Authors' contributions

- Acknowledgements

- Funding statement

on behalf of Professor Guy Genin (Associate Editor) and Pietro Cicuta (Subject Editor)
openscience@royalsociety.org

Associate Editor Comments to Author (Professor Guy Genin):

Comments to the Author:

Thank you for submitting a revision to your initial manuscript. The manuscript is improved, but more work is still needed. The most important requirement is that every conclusion and claim must be backed up by evidence in the paper. A renewed effort to think through the contribution would be helpful as well: nothing needs to be "novel" or "the first" for the paper to be published, but it would be helpful to have a clear statement in the abstract of what you consider the contribution to be.

The reviewer has a checklist of very clear suggestions that I believe will strengthen the manuscript, and I ask you to follow these for the revisions.

Thank you for submitting this very nice work to RSOS!

Reviewer comments to Author:

Reviewer: 3

Comments to the Author(s)

The present manuscript presents a study on the extension rate of individual hyphae in a microfluidic device. Although the authors have collected many data, there are two main problems with the manuscript.

Firstly, the objective of the work is not clear. The abstract seems to indicate that the purpose of the study was to determine if it is acceptable to use these types of devices to study growth of single hypha. However, throughout the manuscript, data are presented indicating that growth in the device is slower than growth on agar and the authors do not address this matter properly. In the previous revision, reviewers indicated that the novelty of the work was unclear and, nevertheless, the authors have not explained their objective in the introduction section. The authors replied "The design of the microfluidic system is indeed not new. The objective of this work was to study the growth of a large number of hyphae in confined conditions, with two different degrees of confinement, in order to better understand why growth velocity varies across scales." However, this information is not in the manuscript and is vague. What do the authors mean by two different degrees of confinement? What scales are they referring to?

Secondly, the authors conclude that there is nutrient depletion in the experiment and this explains why *N. crassa* is more affected by the microchannels, but the medium used in the experiments does not contain a carbon source, therefore, the nutrients used for growth are either inside the hyphae or in the inoculum agar, thus, it is in the inoculum chamber. Consequently, the transport of the mineral medium from the other chambers would unlikely affect growth. This is a main aspect of the discussion that needs to be reviewed. Many arguments made by the authors are not validated by the results.

A detailed list of suggestions is presented below:

- 1) Title: "Microfluidic monitoring of the dynamic behavior of individual hyphae in confined environments". What do you mean by dynamic behavior? Growth rate is mostly constant in the experiments.
- 2) Abstract: "The distributions of growth velocity obtained for each strain were compared to measurements obtained in less confined geometries, either in 2D chambers". Where in the manuscript is the comparison between growth in the channel and in the 2D chambers?
- 3) Figures: it is important that figures are easy to understand based only on the figure itself and its legend, without reading the text. In order to achieve this, I present some suggestions:
 - a) Figure 1: What does "10 x 6 μm " stand for? Are the microchannels cylindrical? Is it 6 or 5.8 μm ?
 - b) Figure 4: Legend should read "Hyphal elongation velocity measured in a microchannel AS a function of the distance y from the entrance of the perfusion chamber". Also, indicate the meaning of each symbol in the legend or in the figure.
 - c) Figure 5b: include a title in y-axis explaining what the percentage means. Also, the expression "Temporal evolution of *T. helices* hyphal growth" used in the legend is vague and does not indicate what data was used to construct the graph. It might help to indicate in the figure how the profile in Figure 5a would be classified. It would be better to use the same terms used in the text in the x-axis: accelerating and decelerating.
 - d) Figure 6a: replace the y-axis title for "colony diameter".
 - e) Figure 6b: Include statistical analysis in (b). In figure legend, explain the symbols v_m and v_c .
 - f) Figure 10: Indicate in the Figure legend what is A, B and C
 - g) Check that all figures are cited in the text.
 - h) Check the numbering of figures and tables that are not in the main text.
- 4) Introduction:

- a) Is there a reason why it is a single paragraph? It is usually separated into 3 to 5 paragraphs, with each paragraph with an idea. Consider fragmenting the text into more paragraphs.
- b) Line 25: "In particular, some concerning organic pollutants can be used as carbon and energy source by strains able to access to these molecules and degrade them." It should be "strains able to access and degrade these molecules"
- c) Line 39: remove the space before the comma "per hour ,"
- d) Line 49: If the device was developed by the authors and is presented for the first time in this paper, it would be better to write "In this study, we propose a microfluidic device to monitor the growth of individual hyphae in microchannels." If the device has been used before, there should be a citation for the previous work.
- e) Line 51: "This set-up aims at mimicking a coned, heterogenous and dynamic environment," The terms "heterogenous" and "dynamic" can mean a lot of different things in terms of devices for microfluid studies. I suggest you explain what kind of studies can be performed with this device.
- f) Line 52: Explain which are the "both strains" used in the work, because it is unclear at this point.
- g) What questions are you trying to answer with your research? At the end of the introduction, present the objective of the paper. In the present version, you only present the purpose of the device.
- 5) Experimental section:
- a) Does the *Talaromyces helices* strain have a strain code? Has this strain been used in previous published works?
- b) Line 73: "The intensity of fluorescence in the chambers was monitored by fluorescence microscopy over 3 h using the FITC filter". Where exactly was fluorescence measured? In the entire chamber? In the microchannels? In the outlet? Perhaps indicate in the figure of the chamber the points of analysis.
- c) Line 76: Inform the composition of the mineral medium or cite a reference to the composition.
- d) Section 2.5: What medium was used? What temperature?
- e) Section 2.5: "maximum and minimum diameter of the colony were measured using ImageJ". For each image, only 1 value of maximum diameter and 1 value of minimum diameter were taken? Why was the median value used instead of the average?
- f) Section 2.5: In "Time-averaged elongation velocities were calculated as", include the symbol v_m .
- g) All image analysis was performed in ImageJ? This is not clearly stated.
- 6) Results
- a) What are the hyphal diameters of the strains tested when grown without confinement? Figure 1b indicates that *N. crassa* is larger than the microchannels. How does it affect growth? These information are given in the discussion but should be presented here.
- b) The data presented in Figure 2 was measured in the inoculation chamber or the microchannels? What does the position y refer to? It would be better to indicate the y positions in the figure of the device.
- c) Line 103: The radius of the perfusion chamber is 1 mm, which is the length the authors used to calculate the characteristic time. The experimental time is shorter, which is expected if the measurement was done in a radius lower than 1 mm and if convective flow occurs in the chamber.
- d) Line 106: "Figure 2(a) reveals that the concentration in the inoculation chamber increased fast in the first 30 min, and reached a plateau for the next two hours.". According to Fig 2a, it actually increased fast in the first 10 min and, based on fig 2b, it reached a plateau in 30 min (the curves in fig 2b overlap after 30 min).
- e) Line 109: "This axial dependency can be explained by convective transport of the fluorescein solution through the microchannel." This sentence is misleading. The authors refer to axial dependency in the chamber caused by flow in the microchannel, but it seems that it refers to

axial dependency in the microchannel. The authors need to improve the added text in lines 109 to 113, because it is only understandable after reading the appendix.

f) Line 122: How does inoculum size affect the velocity of individual hyphae? The metabolic state hypothesis might be valid, in addition to compounds from the media diffusing to the perfusion, but the inoculum size does not affect individual hyphae.

7) Discussion:

a) Line 178: "When all the microchannels contain hyphae, we expect their permeability, and as a consequence the gradient in the inoculation chamber, to vanish" Do you mean the fluid will not reach the inoculation chamber when hyphae are present in the channels? This may be true for *N. crassa*, which seems to occupy the entire channel width, but not for smaller hyphae.

b) "The fact that hyphal elongation velocity does not depend on the channel position y , even when some microchannels remain empty, seems to suggest that this situation occurs even before all the channels are filled with hyphae." I don't think you can conclude this. Since there is no difference in extension rate between the static and the perfusion experiment, I believe the nutrients are not being depleted in the static or perfusion experiment. In fact, the medium used in the perfusion does not contain a carbon source, which is the nutrient whose lack limits growth. The nutrients present in the mineral medium would unlikely be depleted because only small amounts of them are used. Therefore the only effect of the flow would be frictional or removal of biological compounds related to quorum sensing or the mineral medium is washing out the nutrients from the inoculum. In order for you to conclude that there is no flow in the chambers, you should have replaced the circulating buffer with fluorescein after the hyphae had occupied the channels. The nutrients in mineral media are not enough for growth.

c) Line 200: "Variations in elongation velocities between experimental conditions could be explained by a modification of nutrient availability". Which experimental conditions? Which variations in elongation?

d) Line 211: There is a discussion about glucose, but there is no glucose in the medium used. Hyphae are either using carbon from the agar of the inoculum or intracellular reserves.

e) Line 217: it is also limited by the lack of space.

f) "It thus seems that the growth of *N. crassa* is limited by the flux of nutrients diffusing from the perfusion chamber, in static and probably also in dynamic conditions". But there is no carbon source in the perfusion chamber.

g) "Its metabolic rate may already be maximal given the geometric constraints." I don't understand this sentence.

h) Line 224: "Differences in nutrient consumption rate between the two strains might explain why *N. crassa* is more affected than *T. helicus* by nutrient deprivation in microfluidic conditions". I disagree given what is written here. The probable explanation is lack of space, since the diameter of *N. crassa* is larger than the size of the channel.

i) Line 225: "elongation velocity of both strains is so small in comparison with the growth velocity measured at the scale of the colony". The authors can't compare growth in agar with confined growth. In agar, there is substrate (carbon source) available under the colony and O_2 , which are lacking in the device. In addition, there are walls blocking hyphal growth in the device.

j) Figure 7 should be in the results section.

k) You could check if it is the wall effect by measuring linear extension rate after the channels and comparing.

8) Conclusion:

a) Line 280: "This suggests that *N. crassa*, although widely used in the literature, may not be the best experimental model for studies related to soil." I don't understand how you can conclude this.

9) Appendix:

a) What does y represent? Is it position in the microchannels? Please cite the corresponding figure.

b) Figure 8 should be figure A1 because it is in the appendix. Table should be numbered A1.

- c) Line 317: Pressure drop refers to a loss in fluid pressure between 2 points. In “pressure drop in the inoculation chamber”, what are the two points? In “pressure drop in the perfusion chamber”, what are the two points?
- d) Line 318: write constant instead of cst.
- e) Line 331 “This is in agreement with the assumptions made at the beginning of the calculation (unperturbed perfusion how, negligible pressure drop in the inoculation chamber).” Ending the appendix with this statement gives the wrong idea that the assumptions used as starting point are only confirmed by the deduction made, which is not entirely true and which does not prove that the assumptions are true because they were input to the deduction. Whenever possible, assumptions should be proved in the middle of the deduction (e.g. that flow in the channels is insignificant compared to flow in the large chamber).

Reviewer: 4

Comments to the Author(s)

I confirmed the author's revision is proper and recommend to be published in this journal.

Author's Response to Decision Letter for (RSOS-191535.R1)

See Appendix D.

RSOS-191535.R2 (Revision)

Review form: Reviewer 3

Is the manuscript scientifically sound in its present form?

Yes

Are the interpretations and conclusions justified by the results?

No

Is the language acceptable?

Yes

Do you have any ethical concerns with this paper?

No

Have you any concerns about statistical analyses in this paper?

Yes

Recommendation?

Major revision is needed (please make suggestions in comments)

Comments to the Author(s)

The present manuscript presents a study on the extension rate of individual hyphae in a microfluidic device. The objective of the work is clearer in this version and all proposed modifications were included in the new version.

I have a few remaining concerns listed below. The main concern is the characterization of mass transport inside the chambers, reported in Figure 6. This data needs to be further analyzed in order to conclude that convection does occur. It seems that the time required to reach steady-state is too long for the characteristic time calculated by the authors, therefore, they should include other analysis.

A detailed list of suggestions is presented below:

- 1) Abstract: "This study is one of the first to consider the degree of confinement within soil microporosities as a key factor of fungal soil colonization along with physicochemical parameters, notably for bioremediation purposes." Is it really the first? In the introduction, it is stated "By using micro-patterned surfaces to probe the growth of *Candida albicans* in filamentous form, geometric constraints have been shown to affect hyphal growth and morphology [13]." So there have been other works considering the degree of confinement. Also, what are the sizes of pores in the soil microporosities? In the discussion, there should be a detailed argument to this sentence from the abstract.
- 2) Figures:
 - a) Figure 2: review punctuation in the legend
 - b) Figure 4: Was it the same experiment?
 - c) Figure 5b: "In 4 of the channels observed," It is not clear if it is for static experiments or all experiments. It's better to state "In some channels,"
 - d) Figure 6: "In both static and dynamic conditions, the difference between *N. crassa* and *T. helicus* was significant ($p < 0:001$)." Based on the error bars, for the dynamic experiment, there doesn't seem to be a significant difference between both fungi; also, for the same fungus, there aren't significant differences between the treatments. The only significant difference that may occur is between both fungi in static conditions. Please, review this analysis. In addition, vc is explained twice in the legend, it is only necessary in the first appearance, using ().
- 3) Introduction:
 - a) "The objective is to be able to discriminate between phenomena that result from the geometrical constraints and those that are merely a consequence of starvation" How do you achieve this?
- 2) Experimental section:
 - a) "Strains were maintained on malt yeast extract agar medium (MYEA)". Please inform the composition of this medium, since there are variations in the literature.
 - b) Section 2.5: It is written that the plate experiment was done in MYEA. In the reply, it is stated that it is MMG. Verify this information. If the medium used in the chamber experiment and in the plate experiments were different, how can you compare their growth?
- 3) Results
 - a) "The diameter of *N. crassa* hyphae is about 10 μm , in contrast to those of *T. helicus* which are 2 to 5 μm wide depending on their age" Avoid the use of "those", it is better to state "in contrast to the diameter of *T. helicus* which is 2 to 5 μm , depending on their age". Also, with this information, you should mention the size of the channels again, reflecting how hyphal diameter affects growth in the channels.
 - b) The authors use the characteristic time of 10 min, which is the time necessary for fluorescein to be detected in $y=0$, for the validation that convection occurs. However, this is a 1 measurement data, which may have errors in measurement. The authors should use the rest of the data in figure 2 to validate this affirmation. It could be done with mathematical models or equations for the diffusion profile (bell-shape graphs). The profile in figure 2b is unusual because the bell-shape graph should become flat over time, which does not occur. Please, improve the analysis of the data in figure 2 in order to confirm the transport phenomena occurring in your device.
 - c) Line 155: Inoculum size refers to the amount of mycelium in the inoculum, which does not affect the extension rate of individual hyphae, only the number of hyphae formed. If the

authors mean that the amount of nutrient-rich agar is affecting growth, than they have to replace "inoculum size" for "amount of nutrient-rich agar added with the inoculum".

Decision letter (RSOS-191535.R2)

14-Apr-2020

Dear Dr Le Goff:

Manuscript ID RSOS-191535.R2 entitled "Microfluidic monitoring of the growth of individual hyphae in confined environments" which you submitted to Royal Society Open Science, has been reviewed. The comments of the reviewer(s) are included at the bottom of this letter.

Please submit a copy of your revised paper before 07-May-2020. Please note that the revision deadline will expire at 00.00am on this date. If we do not hear from you within this time then it will be assumed that the paper has been withdrawn. In exceptional circumstances, extensions may be possible if agreed with the Editorial Office in advance.

Please note that, under most circumstances, the Editors will only permit one round of revisions - that you have been given this further opportunity to revise is an indication that your work is almost 'over the line', but that you've further work to do so and that the Editors recognise you are making 'good faith' efforts to improve the manuscript. Please be aware that, unless you are able to satisfactorily respond to the requests below, we may not be able to consider the paper further.

If deemed necessary by the Editors, your manuscript will be sent back to one or more of the original reviewers for assessment. If the original reviewers are not available we may invite new reviewers.

- Ethics statement

- Data accessibility

It is a condition of publication that all supporting data are made available either as supplementary information or preferably in a suitable permanent repository. The data accessibility section should state where the article's supporting data can be accessed. This section

should also include details, where possible of where to access other relevant research materials such as statistical tools, protocols, software etc can be accessed. If the data have been deposited in an external repository this section should list the database, accession number and link to the DOI for all data from the article that have been made publicly available. Data sets that have been deposited in an external repository and have a DOI should also be appropriately cited in the manuscript and included in the reference list.

- **Competing interests**

- **Authors' contributions**

- **Acknowledgements**

- **Funding statement**

on behalf of Professor Guy Genin (Associate Editor) and Pietro Cicuta (Subject Editor)
openscience@royalsociety.org

Associate Editor Comments to Author (Professor Guy Genin):

Associate Editor: 1

Comments to the Author:

The manuscript is much strengthened from the original submission. I agree with the reviewer that some further, straightforward analysis is needed to support conclusions drawn from flow fields. I look forward to reading your revised manuscript. Again, thank you for submitting your highest quality work to RSOS.

Reviewer comments to Author:
Reviewer: 3

Comments to the Author(s)

The present manuscript presents a study on the extension rate of individual hyphae in a microfluidic device. The objective of the work is clearer in this version and all proposed modifications were included in the new version.

I have a few remaining concerns listed below. The main concern is the characterization of mass transport inside the chambers, reported in Figure 6. This data needs to be further analyzed in order to conclude that convection does occur. It seems that the time required to reach steady-state is too long for the characteristic time calculated by the authors, therefore, they should include other analysis.

A detailed list of suggestions is presented below:

1) Abstract: "This study is one of the first to consider the degree of confinement within soil microporosities as a key factor of fungal soil colonization along with physicochemical parameters, notably for bioremediation purposes." Is it really the first? In the introduction, it is stated "By using micro-patterned surfaces to probe the growth of *Candida albicans* in filamentous form, geometric constraints have been shown to affect hyphal growth and morphology [13]." So there have been other works considering the degree of confinement. Also, what are the sizes of pores in the soil microporosities? In the discussion, there should be a detailed argument to this sentence from the abstract.

2) Figures:

a) Figure 2: review punctuation in the legend

b) Figure 4: Was it the same experiment?

c) Figure 5b: "In 4 of the channels observed," It is not clear if it is for static experiments or all experiments. It's better to state "In some channels,"

d) Figure 6: "In both static and dynamic conditions, the difference between *N. crassa* and *T. helicus* was significant ($p < 0:001$)." Based on the error bars, for the dynamic experiment, there doesn't seem to be a significant difference between both fungi; also, for the same fungus, there aren't significant differences between the treatments. The only significant difference that may occur is between both fungi in static conditions. Please, review this analysis. In addition, vc is explained twice in the legend, it is only necessary in the first appearance, using ().

3) Introduction:

a) "The objective is to be able to discriminate between phenomena that result from the geometrical constraints and those that are merely a consequence of starvation" How do you achieve this?

2) Experimental section:

a) "Strains were maintained on malt yeast extract agar medium (MYEA)". Please inform the composition of this medium, since there are variations in the literature.

b) Section 2.5: It is written that the plate experiment was done in MYEA. In the reply, it is stated that it is MMG. Verify this information. If the medium used in the chamber experiment and in the plate experiments were different, how can you compare their growth?

3) Results

a) "The diameter of *N. crassa* hyphae is about 10 μm , in contrast to those of *T. helicus* which are 2 to 5 μm wide depending on their age" Avoid the use of "those", it is better to state "in contrast to the diameter of *T. helicus* which is 2 to 5 μm , depending on their age". Also, with this information, you should mention the size of the channels again, reflecting how hyphal diameter affects growth in the channels.

b) The authors use the characteristic time of 10 min, which is the time necessary for fluorescein to be detected in $y=0$, for the validation that convection occurs. However, this is a 1

measurement data, which may have errors in measurement. The authors should use the rest of the data in figure 2 to validate this affirmation. It could be done with mathematical models or equations for the diffusion profile (bell-shape graphs). The profile in figure 2b is unusual because the bell-shape graph should become flat over time, which does not occur. Please, improve the analysis of the data in figure 2 in order to confirm the transport phenomena occurring in your device.

c) Line 155: Inoculum size refers to the amount of mycelium in the inoculum, which does not affect the extension rate of individual hyphae, only the number of hyphae formed. If the authors mean that the amount of nutrient-rich agar is affecting growth, than they have to replace "inoculum size" for "amount of nutrient-rich agar added with the inoculum".

Author's Response to Decision Letter for (RSOS-191535.R2)

See Appendix E.

RSOS-191535.R3 (Revision)

Review form: Reviewer 3

Is the manuscript scientifically sound in its present form?

Yes

Are the interpretations and conclusions justified by the results?

Yes

Is the language acceptable?

Yes

Do you have any ethical concerns with this paper?

No

Have you any concerns about statistical analyses in this paper?

No

Recommendation?

Accept with minor revision (please list in comments)

Comments to the Author(s)

The authors have properly replied to all the comments from the previous revision and the corresponding modifications were made in the text. The only modification that remains is that the explanation for the meaning of the error bars (standard deviation) should be given in the methodology section, since statistical analysis is a method. The discussion about the t-test added in lines 205-206 could remain in section 3.4.

Decision letter (RSOS-191535.R3)

Dear Dr Le Goff:

On behalf of the Editors, I am pleased to inform you that your Manuscript RSOS-191535.R3 entitled "Microfluidic monitoring of the growth of individual hyphae in confined environments" has been accepted for publication in Royal Society Open Science subject to minor revision in accordance with the referee suggestions. Please find the referees' comments at the end of this email.

The reviewers and Subject Editor have recommended publication, but also suggest some minor revisions to your manuscript. Therefore, I invite you to respond to the comments and revise your manuscript.

- Ethics statement

- Data accessibility

<http://datadryad.org/submit?journalID=RSOS&manu=RSOS-191535.R3>

- Competing interests

- Authors' contributions

- Acknowledgements

- Funding statement

Because the schedule for publication is very tight, it is a condition of publication that you submit the revised version of your manuscript before 05-Aug-2020. Please note that the revision deadline will expire at 00.00am on this date. If you do not think you will be able to meet this date please let me know immediately.

Supplementary files will be published alongside the paper on the journal website and posted on the online figshare repository (<https://figshare.com>). The heading and legend provided for each supplementary file during the submission process will be used to create the figshare page, so

please ensure these are accurate and informative so that your files can be found in searches. Files on figshare will be made available approximately one week before the accompanying article so that the supplementary material can be attributed a unique DOI.

Best regards,

on behalf of Professor Guy Genin (Associate Editor) and Pietro Cicuta (Subject Editor)
openscience@royalsociety.org

Associate Editor Comments to Author (Professor Guy Genin):

Congratulations on an outstanding contribution! Many thanks to you for sending this beautiful piece of work to RSOS. The reviewer had a couple of very small suggestions for the final manuscript that would make sense to touch up. If you agree, please make these changes before uploading your final files.

Reviewer comments to Author:

Reviewer: 3
Comments to the Author(s)

The authors have properly replied to all the comments from the previous revision and the corresponding modifications were made in the text. The only modification that remains is that the explanation for the meaning of the error bars (standard deviation) should be given in the methodology section, since statistical analysis is a method. The discussion about the t-test added in lines 205-206 could remain in section 3.4.

Author's Response to Decision Letter for (RSOS-191535.R3)

See Appendix F.

Decision letter (RSOS-191535.R4)

Dear Dr Le Goff,

It is a pleasure to accept your manuscript entitled "Microfluidic monitoring of the growth of individual hyphae in confined environments" in its current form for publication in Royal Society

Open Science. The comments of the reviewer(s) who reviewed your manuscript are included at the foot of this letter.

on behalf of Professor Guy Genin (Associate Editor) and Pietro Cicuta (Subject Editor)
openscience@royalsociety.org

Associate Editor Comments to Author (Professor Guy Genin):

Congratulations on an outstanding contribution to the literature! I really love this paper. Thank you once again for submitting it to Royal Society Open Science!

Appendix A**ROYAL SOCIETY
OPEN SCIENCE****Microfluidic monitoring of the dynamic behavior of
individual hyphae in confined environments**

Journal:	Royal Society Open Science
Manuscript ID	RSOS-191535
Article Type:	Research
Date Submitted by the Author:	19-Sep-2019
Complete List of Authors:	Baranger, Claire; Universite de Technologie de Compiegne Fayeulle, Antoine; Universite de Technologie de Compiegne Le Goff, Anne; Universite de Technologie de Compiegne,
Subject:	microbiology < BIOLOGY, biomechanics < PHYSICS, bioengineering < CROSS-DISCIPLINARY SCIENCES
Keywords:	fungi, microfluidics, growth
Subject Category:	Biochemistry & Biophysics

Author-supplied statements

Relevant information will appear here if provided.

Ethics

Does your article include research that required ethical approval or permits?:

This article does not present research with ethical considerations

Statement (if applicable):

CUST_IF_YES_ETHICS :No data available.

Data

It is a condition of publication that data, code and materials supporting your paper are made publicly available. Does your paper present new data?:

Yes

Statement (if applicable):

The detail of data regarding growth in confined conditions is provided as supplementary material (File LeGoff_ESM_Velocity_Data).

Conflict of interest

I/We declare we have no competing interests

Statement (if applicable):

CUST_STATE_CONFLICT :No data available.

Authors' contributions

This paper has multiple authors and our individual contributions were as below

Statement (if applicable):

CB, AF and ALG designed the research. CB performed the research. CB, AF and ALG analysed the data and wrote the paper.

1 Microfluidic monitoring of the dynamic behavior of individual hyphae 2 in confined environments

Claire Baranger¹, Antoine Fayeulle¹, Anne Le Goff²

¹ TIMR, EA 4297, Université de Technologie de Compiègne, ESCOM, France

² BMBI, UMR 7338 CNRS Université de Technologie de Compiègne, France

September 19, 2019

**Abstract**

The survival of telluric fungi depends on their ability to colonize the soil, where they encounter spatial and
temporal variations of confinement, humidity and nutrient concentration. In this paper we present a microfluidic
device dedicated to the study of fungal growth at the scale of single hyphae, allowing ~~to achieve a~~ fine control
of nutrient and water supply. Time lapse microscopy allowed ~~to monitor~~ the growth of dozens of hyphae
simultaneously. The distributions of growth velocity obtained for each strain were compared to measurements
obtained in less confined geometries, either in 2D chambers or in macroscopic solid culture. For the two strains
used in the study, confinement caused the growth velocity to drop in comparison with unconfined experiments.
In addition, *N. crassa*, a model filamentous fungus, was also limited in its growth by the nutrient supply, while
the microfluidic culture conditions seemed better suited for the telluric fungus *T. helicus*.

1 Introduction

Filamentous fungi play a key role in terrestrial ecosystems, both as a structural component of the soil and through their trophic interactions with other organisms [?]. They are non-motile organisms which colonize their environment through hyphal growth, thereby forming a three-dimensional mycelial network. The filamentous form allows them to grow in confined environments and colonize porous solid matrices, either in the soil or in plant tissues. Hyphae grow from the tip and are able to branch either laterally or apically. This morphology is adapted to their heterotrophic mode of nutrition, enabling fungi to scout their surroundings and absorb nutrients at the growing tips of hyphae. Moreover, hyphae are able to break air-water interfaces, thus crossing air pockets and gaps between soil particles. The methods traditionally used for fungal culture in the lab, i.e. solid agar plates or liquid medium in shake flasks, are not adapted to approach the complex habitat of fungi at the structural and functional levels. Transfer of nutrients, metabolites, genes and viruses typically occur at the submillimetric scale of soil aggregates, and cannot be captured in controlled homogeneous models with little to no confinement [?]. Microfluidic tools have been proposed to study the interactions between microorganisms and their microenvironment, specifically in soils [? ?]. Numerous set-ups have been designed for bacteria [?], but devices intended for fungi and more specifically filamentous fungi remain scarce. In the past 15 years, a few studies have used fungal culture in microfluidic devices to monitor fungal growth patterns in maze-like structures [?], the fungicidal activity of *Bacillus subtilis* on the fungus *Botrytis cinerea* [?], investigate circadian cycles in *Neurospora crassa* [?] and the germination of *Fusarium fujikuroi* [?]. Using micro-patterned surfaces to probe the growth of *Candida albicans* in filamentous form, some geometric constraints have been shown to affect the growth or morphology of hyphae. In this study we developed a new microfluidic device for fungal culture designed to monitor the growth of individual hyphae in microchannels. This set-up aims at mimicking a confined, heterogenous and dynamic environment, while being suited for microscopic observations. Here we chose to investigate the behavior of a soil mould previously cited for its remediation abilities towards hydrocarbons and heavy metals [?], *Talaromyces helicus*, and compare it to the well-studied ascomycete *Neurospora crassa*, a historical model species in genetics, fungal morphogenesis and

40 development [?]. Growth dynamics of both strains were studied with and without ~~the implementation of a~~
41 medium perfusion in the microfluidic chamber and compared with macroscopic growth on solid medium.

14 42 **2 Experimental section**

16 17 43 **2.1 Fungal strains and macro-culture conditions**

A strain of the filamentous fungus *Talaromyces helicus* previously isolated from an industrial contaminated
soil and selected for its PAH biodegradation properties was used for this study. The model species *Neurospora*
*crassa* was obtained from the BCCM/MUCL Agro-food & Environmental Fungal Collection (strain MUCL
041473). Strains were maintained on malt yeast extract agar medium, at 22°C with a 12h-12h light cycle, and
transplanted onto fresh medium every ten days.

31 49 **2.2 Microfabrication**

A design, represented in Figure 1(a), featuring two chambers connected through 50 parallel microchannels was
patterned onto a silicium plate by photolithography, yielding a positive master (Microfactory, Paris). Each
channel is 10 μm wide, 5.8 μm high and 0.5 or 1 mm long. The chambers were 124 μm high. Polydimethyl-
siloxane (PDMS, Sylgard 184, Dow Corning) was cast against the silicium wafer and allowed to cure for at least
2h at 75°C. Inlets were added by punching holes through the PDMS with a biopsy puncher. The negatively
patterned PDMS slab was bound to a glass microscope slide after surface oxidization in a plasma chamber
(Harrick) for 60s. In order to test whether concentration gradients could be implemented in the device, a chip
featuring 1 mm long channels was filled with distilled water, then perfused with a 50 mg/L fluorescein solution
at 9 $\mu\text{L}/\text{min}$ through the distal chamber using a peristaltic pump (Ismatec). The intensity of fluorescence in
the chambers was monitored by fluorescence microscopy over 3 hours using the FITC filter.

Figure 1: (a) Diagram of the microchip pattern. Microchannels for confined hyphal growth are oriented in the x direction and the main chamber is perfused in the y direction. Detail of (b) *T. helicus* and (c) *N. crassa* hyphae growing through $10 \times 6 \mu\text{m}$ microchannels. The hyphal tip and branching points are indicated with white and black arrows respectively.

2.3 On-chip fungal culture

A 2×2 mm mycelium plug was collected from a solid-medium culture of *T. helicus* or *N. crassa* at the edge of the growing colony, and transferred to the inoculation inlet of the chip previously filled with liquid mineral medium [?]. The inlet was then closed with a PDMS plug to prevent drying and the system was placed in a sealed Petri dish in a water saturated atmosphere. The mycelium was allowed to grow at 22°C with a 12h-12h light cycle, until hyphae reached the opening of the channels. After pre-incubation, the system was connected to a peristaltic pump (Ismatec) using PTFE tubing and the distal chamber was perfused with mineral medium at $1.8 \mu\text{L}/\text{min}$. Hyphal growth was monitored in parallel in multiple microchannels over 10 hours. The experiment was repeated 3 times for each species, with and without perfusion with mineral medium. Growth was measured in a total of 280 channels from 15 separate chips, 6 in static conditions and 9 in dynamic conditions.

2.4 Imaging and data analysis

In order to measure colony growth rates on a solid medium, pictures of 3 Petri dishes freshly inoculated with *N. crassa* were taken every 6h for 48h. Similarly, pictures of 3 Petri dishes inoculated with *T. helicus* were taken every day for four days, and then every 2 or 3 days until colonies reached the edge of the dish. For each image,

the maximum and minimum diameter of the colony was measured using ImageJ, and the median diameter was
calculated. Microscopic observations were made using inverted microscope (Leica DMI-8) equipped with a
motorized stage, and images of 5 fields for each chip were taken at a 10× magnification (or 4× for the fluorescein
perfusion) with a camera (Leica DFC 3000G), every 10 minutes for *T. helicus* and every 5 minutes for *N. crassa*,
for at least 10 hours. In each channel, the distance between the channel entrance and the tip of the longest
growing hypha was measured using ImageJ. Channels presenting obvious signs of drying or disbonding from
the glass slide were excluded from analysis, as well as those already filled with a hypha longer than half the
length of the channel at the beginning of the observations. Time-averaged elongation velocities were calculated
as

$$v_m = \frac{x(t_f) - x(t_i)}{t_f - t_i}$$

where t_i and t_f denote the times at which the observation of the hypha started and stopped, and $x(t)$ represents
the position of the hyphal tip at the time t . The maximum observation time $t_f - t_i$ is 10 hours. For hyphae
reaching the extremity of the channel earlier than 10 hours, the last picture with a visible hyphal tip was taken
into account. Spatio-temporal diagrams were generated using the built-in KymoGraph plugin to visualize
elongation rate variations. Image sequences of the 2D growth of branched hyphae were manually thresholded,
then analyzed using the plugin Fractal Analysis. The Box Counting method was used to obtain the fractal
dimension of each image in the sequence. All results are presented as mean \pm standard deviation.

**3 Results**

46 47 79 **3.1 Time-averaged on-chip growth rate**

For both *T. helicus* and *N. crassa*, the mycelium grew until filling the inoculation chamber and reached the
entrance of the microchannels. This initial phase took 3 to 4 days for *T. helicus*, and about 12 hours for
*N. crassa*. Then, individual hyphae started to extend into the microchannels and grew from the inoculation
chamber (top of figure 2(c)) to the perfusion chamber (bottom of figure 2(c)).

We calculated the time-averaged elongation velocity v_m for each individual hypha. All hyphae did not grow at the same **pace**, and figure 2 represents the distribution of elongation rates for *N. crassa* (a) and *T. helicus* (b), in both **static and dynamic conditions**. While time-averaged growth rates appeared to be consistent within a single chip, they varied significantly from one chip to the other. This can be explained by inconsistent inoculum size and metabolic state: from one chip to the next, the amount of biomass and of nutrients inoculated in the chip was slightly variable. In spite of this variability, a clear difference in elongation velocities was found between the two strains. In static conditions, v_m was ~~found to be~~ $190 \pm 46 \mu\text{m/h}$ on average for *N. crassa* and $47 \pm 19 \mu\text{m/h}$ for *T. helicus*.

In dynamic experiments, perfusion with fresh medium was implemented in order to prevent ~~drought~~ and nutrient depletion in the distal chamber. Natural soils are porous media with a high heterogeneity in pore size and distribution, and can be subjected to variations in water saturation and infiltration rates depending on weather conditions. **As a result, average infiltration rates through a given volume of soil can vary in the range of a few millimeters to a few tens of centimeters per hour, while local flow velocities vary greatly depending on pore size and saturation, reaching up to 50 cm/s in cm-wide macropores** [?]. A low flow rate was chosen in order to minimize shear stress on hyphae and medium consumption: with a flow rate of $1.8 \mu\text{L/min}$ and chamber dimensions of $h = 3 \text{ mm}$ by $w = 124 \mu\text{m}$, the average velocity v_f of the fluid in the perfusion chamber was about $80 \mu\text{m/s}$. The associated Reynolds number is defined as $Re = \frac{h v_f}{\nu}$, with $\nu = 10^{-6} \text{ m}^2/\text{s}$ the kinematic viscosity of water. Its value is of the order of 10^{-2} . In this regard, the microfluidic set-up described here is a plausible situation mimicking laminar flow through a mm-sized pore and saturated μm -wide micropores. As the distribution of elongation rates displayed in figure 2 (b) shows, there was no significant difference for *T. helicus* between chips incubated in static conditions and those with a circulation of medium ($v_m = 47 \pm 19 \mu\text{m/h}$ and $45 \pm 16 \mu\text{m/h}$ respectively). Although it did not significantly affect hyphal elongation rates for *T. helicus*, the perfusion efficiently prevented drying and bubble formation, thus allowing for longer observation times.

In dynamic conditions, mycelial growth in *N. crassa* was again faster than in *T. helicus*. Also, in contrast

to *T. helicus*, growth velocities were significantly different for *N. crassa* between static and dynamic conditions:
hyphae grew at $190 \pm 46 \mu\text{m/h}$ on average without circulation of medium and $300 \pm 128 \mu\text{m/h}$ with circulation.

3.2 Temporal variations of on-chip growth rate

Time-lapse images acquired during confined growth were used to build spatio-temporal diagrams such as the
one displayed in figure 2(c). This graphical representation allows for rapid visualization of elongation rate
variations. On such a diagram, growth at a constant rate appears as a straight line, acceleration as a convex
curve and deceleration as a concave curve. On the example shown in figure 2(c), *T. helicus* first grew at a
constant pace, then accelerated. Hyphae sometimes stopped for a while and then resumed their growth. In
*N. crassa*, such pauses were frequently observed when the cross-section of a microchannel was locally reduced
due to a microfabrication defect. Growth of *T. helicus* in static conditions tended to be mostly constant (52%
of all channels) or decelerating (44%), while profiles noted in chips with perfusion were constant (44%) or
accelerating (44%).

3.3 Petri dish growth

*N. crassa* and *T. helicus* were inoculated at the center of Petri dishes and left to grow until reaching the
edge. For both strains, the colony sizes were found to be consistent between replicates and followed a similar
trend: a short lag phase followed by a linear growth phase (Fig. 3(a)). The slope of these curves yields the
radial growth velocity of the colony v_c , for which an almost 10-fold difference was found between the two
strains. The diameter of *T. helicus* colonies after 7 days was $33.4 \pm 0.1 \text{ mm}$, which is consistent with values
found in the literature (25 to 33 mm in 7 days) [? ?]. The colony growth velocity v_c of $17 \mu\text{m}/\text{min}$ for *N.*
*crassa* is consistent with values from the literature [? ?], and close to the value of $24 \mu\text{m}/\text{min}$ reported for
unconstrained hyphal elongation rate of *N. crassa* [? ?]. Growth measured in colony experiments was always
faster than in microfluidic experiments, as indicated by figure 3(b), but the velocity ratio between micro and
macroscale v_m/v_c is smaller for *N. crassa* (Fig. 3(b)).

3.4 Establishment of a concentration gradient in the microchip

Dilution-based gradient generators have been developed for the culture of microalgae in increasing concentra-
tions [?]. Geometries similar to the one presented here, featuring a row of microchannels joining two perfusion
chambers, have been used to implement concentration gradients with co-current or countercurrent perfusions.
Using specific nutrients or signal molecules, such concentration gradients can be used to orientate cell growth
or migration [? ?].

The chip designed for fungal culture was tested without mycelium for the implementation of a gradient. A chip
featuring 1 mm-long channels was perfused with a 50 mg/L fluorescein solution at a flow rate $Q = 9 \mu\text{L}/\text{min}$.
Fluorescence intensity in both chambers was monitored over time through time-lapse microscopy. In the
perfusion chamber, the fluorescence intensity was found to reach its maximal value after 10 minutes of perfusion,
corresponding to a concentration of 50 mg/L.

In the inoculation chamber, the concentration remained well below this level, even after 150 minutes of
perfusion. Supplementary figure ??(b) reveals that the concentration in the inoculation chamber increased fast
in the first 30 minutes, and seemed to reach a plateau for the next two hours. The plateau value depended on
the distance y to the system entrance. The fluorescence intensity decreased with y in a manner that became
time-independent after half an hour of perfusion, as can be seen in supplementary figure ??(c). This axial
dependency is a consequence of convection, as diffusion alone cannot explain variations in the y direction. A
fluid flow helps to transport solutes from the inlet to the inoculation chamber. Because this inoculation chamber
has no outlet, the incoming fluids have to flow out through the main outlet. This means that fluid flows from
the perfusion to the inoculation chamber in some of the straight channels, and in the reverse direction in the
others. A simple one-dimensional model of fluid flow, detailed in the following Appendix, can be used to predict
the flow rate q through the microchannels.

Briefly, we consider the fluid in the inoculation chamber to be at rest. The solution flowing through the
perfusion chamber induces a pressure gradient in the y direction. Close to the entrance, the fluid is pushed
towards the inoculation chamber, from where it then flows back towards the outlet using the downstream

microchannels. We find a linear dependence between the flow rate q and the position y of the microchannel in
the chambers, which explains the shape of the curves seen in supplementary figure ??(c). At the middle of the
chambers (indicated by the yellow triangles), the convective flow rate is null. Close to the channel entrance,
transport towards the inoculation chamber is accelerated by convection (blue circles and green triangles), while
it is slowed down further downstream (red crosses and gray diamonds). The characteristic time for advection
across the channel is 18 minutes in the microchannels at the extremities of the chamber, which is in agreement
with the time scales visible in figure ??. Although the fluid velocity can reach values as large as $55 \mu\text{m/s}$,
the dimensions of the channels are so small that q is inferior to Q by at least six orders of magnitude, which
explains why the fluorescein concentration in the perfusion chamber remains unchanged: even close to the
outlet, the incoming stream of dilute solution from the microchannel is drowned into an overwhelming flow
of pure solution arriving from the inlet. At such a flow rate, removing all the fluorescein-poor fluid from the
inoculation chamber would take between 9 and 10 days. Finally, a concentration gradient can be achieved by
perfusing one of both chambers, resulting in a polarized environment that can be maintained in a relatively
stable state over several hours.

36 37 38 171 4 Discussion

39 40 41 172 4.1 Water and nutrient transport

For microfluidic fungal culture experiments, the total flow rate was decreased to $Q = 1.8 \mu\text{L/min}$, resulting
in a maximal velocity of $11 \mu\text{m/s}$, which is still larger than the velocity of growing hyphae. It would be
easy to imagine that such a flow would slow down hyphae growing in the opposite direction, close to the
system entrance, and accelerate hyphae growing the the microchannels closer to the system outlet. We did not
notice any influence of y on the hyphal growth velocity v_m . This may be due to the fact that, as microchannels
partially obstructed by hyphae become much more resistive, the flow pattern observed in fluorescein experiments
rearranges. The resulting situation is complex because fungi themselves have developed numerous strategies

to adapt to their ever-changing environment, some of which involve fluid mechanics challenges [?]. Hyphal
growth is associated with an intracellular water flux from the mycelium core towards the apex [?]. In *N. crassa*,
it has been demonstrated that this mass flow is driven by osmotic pressure gradients [?]. In our system, to
replenish the inoculation chamber of the water that feeds growing hyphae, a water flow in the opposite direction
is necessary. This water transfer could be extracellular and take place in empty channels, if there are any left,
or in the occupied channels in the interstitial space between the hyphae and the channel walls. Else, hyphae
that were the first to cross the microchannels could tap water from the perfusion chamber and be home to
intracellular reverse water flow. In some of our experiments, all the microchannels were occupied by fungi and
new hyphae kept growing between the two chambers. The presence of fungi-free channels is thus not necessary.
We can affirm that water transport takes place through fungi-crowded microchannels, either around and/or
through the cells. Time-lapse observations did not provide any evidence of vesicle transport in the reverse
direction, but only observations with an increased time resolution could totally rule out the intracellular water
flow hypothesis, as the average value reported for protoplasmic flows in *N. crassa*, $2 \mu\text{m/s}$, is too fast to be
observed with the time interval that was used in our experiments.

Hyphal growth requires not only water, but also nutrients: glucose sensing has indeed been demonstrated to
be one of the main driving parameters for hyphal growth notably for *N. crassa* [?] along with other nutrients
such as ammonium, aminoacids, phosphates or trace elements, and gases such as oxygen, carbon dioxide and
ammonia [? ? ?]. The diffusion coefficient of glucose ($0.6 \cdot 10^{-11} \text{ m}^2 \cdot \text{s}^{-1}$) is $100 \times$ smaller than that of fluorescein
($0.64 \cdot 10^{-9} \text{ m}^2 \cdot \text{s}^{-1}$), which means diffusion is $100 \times$ slower for glucose than for fluorescein. As a consequence,
glucose transport in the mycelial growth experiments will be purely convective, at least at the beginning of
the perfusion. This could explain the slow growth of *N. crassa* in static conditions. Solute transport has been
discussed in the previous paragraph in a simplified situation in the absence of fungi. In actual hyphal growth
experiments, convection and diffusion are not the only physical phenomena involved in nutrient mass balance,
because fungal growth also leads to nutrient consumption. Since hyphal growth is not synchronized between
channels, this effect could lead to concentration disparities between channels. These variations could be linked

to regulatory processes controlling the local behavior and intrahyphal nutrient redistribution depending on
signals coming from other parts of the mycelium [?]. Hyphae may be equally capable of growing with the flow
or against the flow, but the response to a flow depends on the strain. The apical extension velocity of *T. helicus*
is almost insensitive to flow. Nutrient availability thus does not appear to be the main limiting factor for the
growth of *T. helicus* in these conditions, and its metabolic rate may already be maximal given the geometric
constraints and limited access to oxygen. The strain's relatively slow growth seems associated with a moderate
consumption of nutrients and its needs are met with the initial amount of nutrient in the chamber, in contrast
to *N. crassa* whose growth rate requires continuous feeding through perfusion.

24 25 213 4.2 Effect of confinement

*N. crassa* and *T. helicus* do not only differ by their metabolism but also by their morphology, as illustrated by
figures 1(b) and (c). *N. crassa* hyphae display a diameter of about 10 μm , in contrast to those of *T. helicus*
which are 2 to 5 μm wide depending on their age. As a consequence, single hyphae of *N. crassa* tightly fit into
the microchannels, while each microchannel can contain multiple hyphae of *T. helicus*. This means that the
degree of confinement differ for both species. Although the growth speed of *N. crassa* was larger than that of
*T. helicus* in all the experiments, confinement had more impact on *N. crassa*, as observed in figure 3(b): the
ratio between confined hyphal growth rate and colony growth rate is larger for *T. helicus* than for *N. crassa*.
This action of confinement could be due to the detection of the microchannel surfaces by hyphae membranes,
since several types of mechanosensitive receptors have been shown to control fungal morphogenesis including
hyphal growth [?]. Notably, for *N. crassa*, thigmotropism has been proposed to be due to a putative calcium
channel protein responding to mechanical stresses at the apex and triggering changes in the polarity machinery
at the hyphal tip [?]. In addition, the confinement degree could also influence growth dynamics by impairing
nutrient transport in the liquid layer surrounding the growing hyphae in the microchannel.

For *T. helicus*, branching events were observed in the straight channels and usually associated with a brief
pause or a decrease in elongation rate. This arrest of mycelial growth prior to branching events was also

observed ~~in experiments studying~~ the unconfined growth of single hyphae [?]. We did not observe branching
of *N. crassa* during its growth in the confined channels, in spite of the fact that the channels were longer than
the strain's branching distance [?], probably because confinement was too high. Confinement has been shown
to influence the size of reproductive units and thus genetic diversity in mycelia [?].

In microchannels where fungal growth started late, we sometimes observed the simultaneous growth of two
hyphae in opposite directions, one of them originating from the perfusion chamber that had been reached by
faster neighbors. In this case, a pause was observed long before the two hyphae were in contact, then one of
them resumed its growth towards the other one. This could be an illustration of a quorum sensing phenomenon
between hyphae, which is poorly understood and studied in fungi, but has been shown to exist in *N. crassa*
and to control fungal growth in other species [?]. More precisely, quorum sensing have already been shown to
play an important role in fungi to detect cell density of the entire population in order to coordinate a beneficial
response to prevent competition for nutrients [?]. The confinement degree according to the hyphal diameter
could thus influence the accumulation of the signal molecule in the channel and the influence on the growth of
other hyphae.

The presence of an hypha in a channel did not, however, automatically prevent new hyphae coming from
the inoculation chamber to colonize the channel. Simultaneous growth of multiple hyphae happened more
frequently with *T. helicus* than with *N. crassa*. The second hypha generally grew more slowly than the first
one, but the amount by which it was slowed down differed depending on the strain: the velocity decreased by
16 % in *T. helicus* and 69 % in *N. crassa*. This again illustrates the fact that *N. crassa*, being more confined
in the channel than *T. helicus*, is also more sensitive to an additional confinement.

Although the main focus of this work was to study the growth of single hyphae in the microchannels, images
acquired in the perfusion chamber, such as the one displayed in figure 4(a), can be used to characterize the
ability of fungi to colonize a soil. The tip velocity of single hyphae is not sufficient to predict this capacity, as a
given total hyphal length can be associated with a diversity of mycelial morphologies [?]. Fractal geometry has
been used for decades to describe fungal growth. The fractal dimension FD characterizing mycelial growth on

a surface is a number between 1 and 2. Automated image analysis can be performed as long as the mycelium is
not too dense, either in a dilute suspension of mycelial fragments [?], at the early stages of pellet formation [?
256], or at the boundary of growing colonies [?] Figure 4(b) shows how pictures obtained by time-lapse microscopy
were post-treated to extract the mycelial outline. 4(c) presents the time evolution of fractal dimension. As
hyphae started to grow out of the microchannels, the fractal dimension was close to 1 and increased over 10
259 hours to reach a value between 1.4 and 1.5. To the best of our knowledge, no values of the fractal dimension
characterizing the growth of *T. helicus* are available in the literature, but our experimental values are consistent
with those reported for other strains. The surface occupied by the mycelium increased over time, as indicated
by figure 4(d). It did so at a faster rate in the first 10 hours because the network was getting denser and at the
same time expanding its domain. After 10 hours, the fractal dimension was constant and the occupied surface
grew at a constant, slower pace.

5 Conclusion

The culture of filamentous fungi was achieved in a microfluidic device featuring two chambers connected by
parallel microchannels, in presence or absence of a perfusion. This device allows for single cell scale monitoring
and parallelization of the observations: up to 33 single hyphae could be measured simultaneously on the same
chip. Besides, this technique of microfluidic culture with a continuous flow of fresh medium avoids drying and
nutrient depletion, enabling longer culture times. *N. crassa* colonies were found to grow ten times faster than
*T. helicus* on agar, while hyphal elongation rates in microchannels were 4-fold higher for *N. crassa* than *T.*
*helicus* in static conditions, and 7-fold higher in dynamic conditions. Our system was well adapted to create
a stable environment for the slow-growing strain *T. helicus*, which we assume to be representative for telluric
fungi. We also used it with the model strain *N. crassa*, widely used in the literature, which grows so fast
that it is limited by the low nutrient supply in confined conditions. This system could be used in the future
for fundamental studies, for instance to investigate physiological phenomena at the hyphal apex or as tool

for applied microbiology, for example to extract experimental values for the parameters involved in models
of fungal growth in silico [?]. Our device contains compartments well suited for the study of the growth
of single hyphae, and others where the network growth dynamics can be measured, and allows to study the
influence of confinement on fungal growth. In the specific case of the depolluting fungus *T. helicus* studied
here, it could be useful to understand fungal interactions with pollutants within soil matrices at a microscale.
Similar setups could be used for the culture of other filamentous organisms such as actinobacteria or microalgae.
Further research is needed to implement more accurate experimental models to study soil microorganisms and
microbial communities, and soil-on-chip approaches could be one of the answers to this lack.

**6 Acknowledgements**

CB benefited from a doctoral grant from the French Ministry of Research. The MycoFlu project has been
supported by grants from the CNRS EC2CO program and Sorbonne Universités Emergence program. The
authors would like to thank Xue Sun, Marie Valmori, Roxane Valentin and Théo Guillerm for their help with
the preliminary experiments.

**References**

[1] Henrik Bruus. *Theoretical Microfluidics*. Oxford University Press, 2008.

Figure 2: Distribution of hyphal elongation rates in microchannels, with and without perfusion for (a) *N. crassa* and (b) *T. helicus*. The average elongation rate, standard deviation and size of each sample are displayed in labels. Student's t-test was performed for each strain and the p-value is labeled on each graph. (c) Detail of the hyphal elongation of *T. helicus* in one microchannel over 15 hours (growth from left to right). (a) Empty channel at the initial state. (b) Spatio-temporal diagram of hyphal length vs time. Black lines were drawn to outline linear elongation phases. Slope changes and a full elongation stop are indicated by black arrows. (c) channel filled with hyphae at the final state.

Figure 3: (a) Colony diameter over time for *N. crassa* and *T. helicus* grown on MYEA agar (average and standard deviation for triplicates). Labels indicate the colony radial growth velocities v_c computed from the slopes of the linear fits. (b) Ratio between velocities measured at microscopic scale in confined conditions and velocity measured at the scale of the colony.

Figure 4: (a) Micrograph showing *T. helicus* hyphae growing out of the straight channel. Branching is visible. (b) Time series showing the progression of 2D growth in the chamber. Image threshold is adjusted to automatize measurements. (c) Fractal dimension as a function of time. (d) Surface coverage as a function of time.

Appendix B

Microfluidic monitoring of the dynamic behavior of individual hyphae in confined environments

Baranger et al present a manuscript that details a microfluidic system for monitoring the dynamic behavior of individual hyphae in confined environments. They describe the growth of two fungi, *T. helicus* and *N. crassa*, and claim that medium perfusion influences the rate of fungal growth (compared to static conditions). The emphasis of the manuscript appears to be on measuring the growth of the two fungi in the device. However, it is not clear what the novelty of this work is based on the described experiments. My major concerns are detailed below, as well as minor changes that need to be made.

Major revisions

- The title of the manuscript itself does not suggest that the work is novel, as there are several manuscripts (which the authors refer to in their introduction) that already allow this capacity. The statements made in Lines 34-37 are NOT novel. What does this new microfluidic device offer over those already published? This is a serious point that needs to be addressed throughout the manuscript to provide a clear focus for the reader.
- Upon reading further, I was excited to learn that the authors had mentioned the potential to subject growing hyphae to concentration gradients, which is novel. However, unfortunately, I could not find any data in which they had shown that concentration gradients could be set up in the presence of hyphae, or any proof-of-concept experiments showing how hyphae in this device could respond differently to a concentration gradient. Moreover, the authors discuss the fact that in their device (without fungi) there will be a flow in two opposing directions through the microchannels (required to set up the concentration gradient). However, if there are hyphae growing through these microchannels they will effectively plug them (to different extents, depending on the number of hyphae that squeeze through these narrow constriction channels) which will disrupt (and remove) the concentration gradient. Thus, I have major concerns regarding the feasibility of creating concentration gradients in the presence of hyphae in this device and I am not sure what is really novel about this manuscript. At present, the novelty of the manuscript is very low.
- I have a concern that the hyphae at the start of the perfusion channel will experience a different fluidic environment to those at the end of the perfusion channel. Can the authors prove that there is no effect of hyphal position on growth rate, i.e. did they analyse their data to see if there is any relationship between hyphal position in the channel (y direction, as indicated in Figure 1a) and growth rate? The authors mention in lines 177 that they “did not notice” any influence of y on growth rate, but this important control needs to be backed up with data. If they can prove this, then this would be one angle that would strengthen the novelty of the manuscript.

Minor comments

- Line 31 error: *Coprinopsis* not *Botrytis*

- Line 34 error: developed not developed
- Line 47: should read “12h-12h light-dark cycle” (or equivalent)
- Figure 1 (a) needs to be more detailed. Are the circles inlets, outlets? These need to be labelled. A “to-scale” mask design should be provided as supplementary information. Although the microchannel length of 500 μm is indicated, a proper scale bar should be included to ensure that the drawing is to scale and therefore a true representation of the device.
- Figure 1 (a): Please include a photograph of the microfluidic setup and label accordingly.
- Line 52: why are the microchannels 0.5 or 1 mm long? Is this justified somewhere?
- Line 54: how big is the biopsy puncher? Please specify diameter. From Figure 1a, it seems as if this could be several mm in diameter.
- Lines 56-59: Is this data displayed somewhere?
- Are the devices prepared with liquid medium or water before each experiment (regardless if there is perfusion or not)? Please elaborate.
- Line 66: How do you connect the tubing to the PDMS device? No details are provided.
- Lines between 70 and 71 (for some reason no line number here in the version I am viewing): disbonding is not a word, please reword
- Line 80: English needs rewording
- Figure 2 legend: please reword/edit. It refers to (a), (b), (c) and then again to (a), (b), (c). Not clear.
- Figure 2 (c) is not so clear. Is this a kymograph? Please indicate time increments on the arrow bar on the left hand side (time axis, I assume this is 15 h total). It would be nice to indicate the image of the hypha (taken at 9h) with a dotted line on the kymograph.
- Line 86-87: is this data shown in a figure? If not, please show in a Supplementary figure.
- Line 106: you seem to have problems with drying of the channel. Was this just for *T. helices*? If yes, why do you think this might be?
- I would like to see some representative supplementary videos showing the growth of each fungus in the channels, for each condition (flow and no flow).
- Line 115-6: Really? It looks like the gradient is constant (i.e. growth is constant) and then stops.
- Lines 118-120: Where is the data to support these claims
- Line 123: add space
- It seems the hyphae in devices grow 2x and 5x slower compared to the petri dish. Why is this i.e. why are the ratios not 1?
- Line 140: which chambers? Not clear.
- Supplementary Figure S1. It is not clear where the data in (b) and (c) was collected (relative to the sketch in (a)). Needs clarification. Also, is this data only from one replicate? The authors should use a programme like COMSOL to model the flows in this device.
- Lines 168-170: where is the data to prove this?
- Line 189: where is the data to prove this? Or are you hypothesizing that this is happening?
- Figure 4: are the processed images in (b) supposed to be based on the microscope image in (a)? Scale bars are missing and I do not see how the first tile in (b) is linked to (a). A major comment is that I would like to see some control

measurements to prove how accurately your image processing technique represents the actual fungal mycelium.

Appendix C

Compiègne, November 10th, 2019

Dear Editor,

Please find enclosed a revised version of our manuscript. We thank the referees for their careful reading of the manuscript and their valuable insight. We agree with all their comments and we have done our best to meet their expectations in the revised manuscript. Briefly, the motivations for studying *T. helicus* have been developed in the Introduction. Misleading statements about concentration gradients have been deleted, new figures have been added to support the claims in the Results section, and the Discussion section has been completely rewritten. Major modifications in the manuscript have been highlighted in color. Please find below our detailed answers to the referee's comments. We hope that you will now find our manuscript suitable for publication in Royal Society Open Science.

On behalf of the co-authors,

Anne Le Goff

Reviewers' Comments to Author:

Reviewer: 1

Comments to the Author(s)

The manuscript deals with the growth of two filamentous fungi in a microfluidic device: *Talaromyces helicus* (which can be used in bioremediation of hydrocarbons and heavy metals) and *Neurospora crassa* (a "model" filamentous fungus). Unfortunately, it is not very clear how the manuscript advances scientific knowledge.

Do the advances have something to do with the design of the microfluidic device? If so, the advantages of the device were not properly discussed in the context of the literature regarding the growth of filamentous fungi in microfluidic devices. Do the advances relate to the particular results that were obtained with the two fungi? If so, is not clear whether the comparison between *T. helicus* and *N. crassa* serves any useful purpose.

The design of the microfluidic system is indeed not new. The objective of this work was to study the growth of a large number of hyphae in confined conditions, with two different degrees of confinement, in order to better understand why growth velocity varies across scales.

Additionally, although the authors suggest that their microfluidic device can mimic the confinement that occurs within soils, it is not clear whether their results have any real implications for understanding fungal growth in soils. Part of the problem is that the discussion is quite speculative and imprecise.

The scope of the discussion has been redefined in order to make it more precise. We now focus on explaining the reduced growth velocity measured in confined conditions. There are several reasons why growth is slowed down in microsystem: one of them is the limited nutrient supply, which account for slow growth in *N. crassa* but not in *T. helicus*, which grows as slowly in dynamic conditions as in static conditions. Confinement also modifies branching distance and hyphae mutual interaction, which affect the apparent growth rate at the scale of the colony.

In many places, it is not very clear what the authors are trying to say. The discussion needs to be completely rewritten – the authors should limit themselves to conclusions that are supported by the data and also need to write in a clearer, more precise manner.

The Discussion section has been entirely rewritten and we hope that the text is now more clear.

In general, the paper needs to make it explicitly clear exactly what is the advancement of scientific knowledge that it contributes - in the context of the current literature in the area of that advancement.

Please note that I made some comments directly onto a PDF of the manuscript, which I uploaded onto the system. However, the authors need to realize that it is not simply a case of addressing these comments one by one - I could have made many, many more comments than I did.

The remarks of the referee have been addressed in the revised version of the manuscript.

Reviewer: 2

- The title of the manuscript itself does not suggest that the work is novel, as there are several manuscripts (which the authors refer to in their introduction) that already allow this capacity. The statements made in Lines 34-37 are NOT novel. What does this new microfluidic device offer over those already published? This is a serious point that needs to be addressed throughout the manuscript to provide a clear focus for the reader.

The design of the microfluidic system is indeed not new. The objective of this work was to study the growth of a large number of hyphae in confined conditions, with two different degrees of confinement, in order to better understand why growth velocity varies across scales.

- Upon reading further, I was excited to learn that the authors had mentioned the potential to subject growing hyphae to concentration gradients, which is novel. However, unfortunately, I could not find any data in which they had shown that concentration gradients could be set up in the presence of hyphae, or any proof- of-concept experiments showing how hyphae in this device could respond differently to a concentration gradient. Moreover, the authors discuss the fact that in their device (without fungi) there will be a flow in two opposing directions through the microchannels (required to set up the concentration gradient). However, if there are hyphae growing through these microchannels they will effectively plug them (to different extents, depending on the number of hyphae that squeeze through these narrow constriction channels) which will disrupt (and remove) the concentration gradient. Thus, I have major concerns regarding the feasibility of creating concentration gradients in the presence of hyphae in this device and I am not sure what is really novel about this manuscript. At present, the novelty of the manuscript is very low.

The referee is right: growing hyphae plug the microchannel, thus disrupting the flows that generate the concentration gradient in absence of fungi. There are no gradients in our systems with fungi, which is why all hyphae are analyzed as a homogeneous population. We agree that the presentation of these results in the original manuscript was misleading. In the revised manuscript, we put more emphasis on the fact that fungal development in microsystems took place without such gradients.

- I have a concern that the hyphae at the start of the perfusion channel will experience a different fluidic environment to those at the end of the perfusion channel. Can the authors prove that there is no effect of hyphal position on growth rate, i.e. did they analyse their data to see if there is any relationship between hyphal position in the channel (y direction, as indicated in Figure 1a) and growth rate? The authors mention in lines 177 that they “did not notice” any influence of y on growth rate, but this important control needs to be backed up with data. If they can prove this, then this would be one angle that would strengthen the novelty of the manuscript.

A new figure was added in the revised manuscript, showing that the variability in elongation velocity could not be explained by the position of the microchannel in the system, neither for *T. helicus* nor for *N. crassa*. Thus, our system can be used as an array of 50 equivalent microchannels. The Results and Conclusion sections have been modified to highlight this fact.

Minor comments

Figure 1 (a) needs to be more detailed. Are the circles inlets, outlets? These need to be labelled. A “to-scale” mask design should be provided as supplementary information. Although the microchannel length of 500 μm is indicated, a proper scale bar should be included to ensure that the drawing is to scale and therefore a true representation of the device.

The figure has been modified, with labels indicating the inlets/outlet, and the chambers are drawn to scale.

- Figure 1 (a): Please include a photograph of the microfluidic setup and label accordingly. A photograph of the microfluidic setup is provided as supplementary information.

- Line 52: why are the microchannels 0.5 or 1 mm long? Is this justified somewhere?

The length was chosen with respect to the microscope field of view. A sentence about this has been added in the Materials & Methods section.

- Line 54: how big is the biopsy puncher? Please specify diameter. From Figure 1a, it seems as if this could be several mm in diameter.

We used two different punchers: one for the perfusion inlet and outlet (1 mm) and one for the inoculation inlet (2 mm). A sentence about this has been added in the Materials & Methods section.

- Lines 56-59: Is this data displayed somewhere?

This experiment corresponds to the data discussed in section 3.1 of the revised manuscript.

- Are the devices prepared with liquid medium or water before each experiment (regardless if there is perfusion or not)? Please elaborate.

All microsystems are prepared in the same manner: they are filled with mineral medium, seeded with mycelium and pre-incubated until hyphae reach the entrance of the microchannels. Then, the pump is turned on for the systems to be cultured in dynamic conditions. A sentence has been added in the Materials & Methods section.

- Line 66: How do you connect the tubing to the PDMS device? No details are provided.

The PTFE tubing is directly inserted in the PDMS inlet/outlet holes. A sentence was added in the Materials & Methods section and the circuit can be seen in Supplementary Figure 9 of the revised manuscript.

- Figure 2 legend: please reword/edit. It refers to (a), (b), (c) and then again to (a), (b), (c). Not clear.

The legend has been rewritten.

- Figure 2 (c) is not so clear. Is this a kymograph? Please indicate time increments on the arrow bar on the left hand side (time axis, I assume this is 15 h total). It would be nice to indicate the image of the hypha (taken at 9h) with a dotted line on the kymograph.

The error in the legend has been fixed. The total length of the analyzed sequence is 9 h, so the two micrographs represent the initial and final state.

- Line 86-87: is this data shown in a figure? If not, please show in a Supplementary figure.

The data have been added in a Supplementary Figure.

- Line 106: you seem to have problems with drying of the channel. Was this just for *T. helices*? If yes, why do you think this might be?

Drying occurred in static conditions when inoculations inlets were not plugged in a perfectly tight way. This happened indeed more often with *T. helicus*, because its slow growth imposes longer experiments.

- I would like to see some representative supplementary videos showing the growth of each fungus in the channels, for each condition (flow and no flow).

Videos have been added as supplementary material.

- Line 115-6: Really? It looks like the gradient is constant (i.e. growth is constant) and then stops.

Black lines highlighting the slope change have been added. The line is indeed interrupted, not because the growth stops, but because the hyphal tip leaves the field of view before the end of the 9 hours.

- Lines 118-120: Where is the data to support these claims

These data are now plotted in figure 5(b).

- It seems the hyphae in devices grow 2x and 5x slower compared to the petri dish.

Why is this i.e. why are the ratios not 1?

There are several reasons why growth is slowed down in microsystem: one of them is the limited nutrient supply, which account for slow growth in *N. crassa* but not in *T. helicus*, which grows as slowly in dynamic conditions as in static conditions. Confinement also modifies branching distance and hyphae mutual interaction, which affect the apparent growth rate at the scale of the colony. These points are raised in the Discussion section.

- Line 140: which chambers? Not clear.

Measurements were made in inoculation and perfusion chambers. The sentence has been modified.

- Supplementary Figure S1. It is not clear where the data in (b) and (c) was collected (relative to the sketch in (a)). Needs clarification. Also, is this data only from one replicate? The authors should use a programme like COMSOL to model the flows in this device.

Measurements were made in the inoculation chamber, at different distances y , close to the inlets of the microchannels. This precision has been added to the text. We did not perform numerical simulation of the flow, as modeling the flow in presence of fungi would require detailed knowledge of fungal shape and motion.

- Lines 168-170: where is the data to prove this?

This sentence was supported by the fact that the axial dependency of the concentration in the inoculation chamber was stable for several hours. It has been removed from the revised version, as it wrongly conveyed the idea that such a gradient was also present in experiments with fungi.

- Line 189: where is the data to prove this? Or are you hypothesizing that this is happening?

The message in this sentence was that hyphae manage to grow from the inoculation towards the perfusion chamber, even when all channels were crowded with fungi. The paragraph has been rewritten.

- Figure 4: are the processed images in (b) supposed to based on the microscope image in (a)? Scale bars are missing and I do not see how the first tile in (b) is linked to (a). A major comment is that I would like to see some control measurements to prove how accurately your image processing technique represents the actual fungal mycelium.

As we now explain in the Materials & Methods section, the length of microchannels is chosen to allow for time-lapse observation of the whole microchannels. Figure 4(a) (now 7(a)) is just a snapshot taken at the end of such a time-lapse. After this date, microchannels are full and we do not expect much information from this region, which is why we focus on further fields of view in figure 4(b) (now 7(b)). Scale bars have been added. The image processing routine has been performed several times, varying the parameters such as threshold and contrast, with no significant effect on the fractal dimension or on the shape of the curves displayed in panels (c) and (d). The absolute value of the surface coverage is however very sensitive to this protocol, and therefore should be handled with care.

Dear Editor,

Please find enclosed a second revision of our manuscript. We thank the referees for their time and relevant feedback, which helped us improve the manuscript.

Briefly, we rephrased the end of the Introduction section in order to clarify the objectives of the work. We added experimental details on the composition of medium in order to better support the claims that are made in the discussion. The presence of glucose in the culture medium was not explicitly mentioned in the previous version and the referee was right to point that the whole discussion about nutrient was pointless if the culture medium did not contain any glucose. We also made our best to take into account all the remarks made by the referee.

Modifications in the manuscript have been highlighted in color. Please find below our detailed answers to the referee's comments. We thank you sincerely for your support and hope that you will now find our manuscript suitable for publication in *Royal Society Open Science*.

On behalf of the co-authors,

Anne Le Goff

Response to Reviewer #3

1) Title: "Microfluidic monitoring of the dynamic behavior of individual hyphae in confined environments". What do you mean by dynamic behavior? Growth rate is mostly constant in the experiments. We replaced *dynamic behavior* with *growth* to improve clarity of the title.

2) Abstract: "The distributions of growth velocity obtained for each strain were compared to measurements obtained in less confined geometries, either in 2D chambers".

The referee is right, there is a quantitative comparison between growth in microchannels and growth in Petri dishes, while the comparison between 1D (microchannels) and 2D (microfluidic chambers) growth in confined environment is more qualitative and mostly focuses on branching. The abstract has been modified.

3) Figures: it is important that figures are easy to understand based only on the figure itself and its legend, without reading the text. In order to achieve this, I present some suggestions:

a) Figure 1: What does "10 x 6 μm " stand for? Are the microchannels cylindrical? Is it 6 or 5.8 μm ? Microchannels have a rectangular 10 x 5.8 μm cross-section. The legend has been corrected.

b) Figure 4: Legend should read "Hyphal elongation velocity measured in a microchannel AS a function of the distance y from the entrance of the perfusion chamber". Also, indicate the meaning of each symbol in the legend or in the figure.

The legend of figure 4 has been corrected.

c) Figure 5b: include a title in y-axis explaining what the percentage means. Also, the expression "Temporal evolution of *T. helices* hyphal growth" used in the legend is vague and does not indicate what data was used to construct the graph. It might help to indicate in the figure how the profile in Figure 5a would be classified. It would be better to use the same terms used in the text in the x-axis: accelerating and decelerating.

The graph has been replaced with two pie-charts in order to better express the fact that we are interested in comparing the distribution of behaviors (acceleration, constant velocity or deceleration) among hyphae cultured in static or in dynamic conditions. The labels have been modified in agreement with the referee's advice.

d) Figure 6a: replace the y-axis title for "colony diameter".

The y-label of figure 6a has been corrected.

e) Figure 6b: Include statistical analysis in (b). In figure legend, explain the symbols v_m and v_c .

The figure 6b and its legend have been modified

f) Figure 10: Indicate in the Figure legend what is A, B and C

A, B and C represent three separate experiments. This information has been added to the legend of figure 10 (now B2).

g) Check that all figures are cited in the text.

All figures are cited in the text.

h) Check the numbering of figures and tables that are not in the main text.

The numbering of figures and table in the appendix has been modified.

4) Introduction:

a) Is there a reason why it is a single paragraph? It is usually separated into 3 to 5 paragraphs, with each paragraph with an idea.

The text has been fragmented into 5 paragraphs.

b) Line 25: "In particular, some concerning organic pollutants can be used as carbon and energy source by strains able to access to these molecules and degrade them."

The sentence has been changed to "strains able to access and degrade these molecules", following the referee's suggestion.

c) Line 39: remove the space before the comma "per hour ,"

The space has been removed.

d) Line 49: If the device was developed by the authors and is presented for the first time in this paper, it would be better to write "In this study, we propose a microfluidic device to monitor the growth of individual hyphae in microchannels." If the device has been used before, there should be a citation for the previous work.

The sentence has been modified to indicate that the device is here used for the first time.

e) Line 51: "This set-up aims at mimicking a conned, heterogenous and dynamic environment," The terms "heterogenous" and "dynamic" can mean a lot of different things in terms of devices for microfluid studies. I suggest you explain what kind of studies can be performed with this device.

By heterogeneous we understand the fact hyphae in this system interact not only with the liquid but also with the solid phase (microchannel walls), and that the geometry allows to generate gradients of nutrient.

Experiments performed with culture medium flowing through the perfusion chamber are called dynamic. The sentence has been modified to clarify these points.

f) Line 52: Explain which are the "both strains" used in the work, because it is unclear at this point.

The sentence has been modified to explicitly name the two strains.

g) What questions are you trying to answer with your research? At the end of the introduction, present the objective of the paper. In the present version, you only present the purpose of the device.

The end of the introduction has been rewritten to better present the objective of the paper. Briefly, most of the studies of fungal growth in microfluidic systems that can be found in the literature were performed in static conditions. In our study, we seek to combine geometric constraints at the scale of the single hypha and a liquid flow. This allows us to discuss whether phenomena observed under microscale confinement result from the geometrical constraints or are simply caused by starvation, which occurs easily in microfluidic chambers due to the small dead volumes in these devices.

5) Experimental section:

a) Does the *Talaromyces helices* strain have a strain code? Has this strain been used in previous published works?

This strain was isolated from a contaminated soil and comes from the laboratory collection. This is now indicated in the manuscript.

b) Line 73: "The intensity of fluorescence in the chambers was monitored by fluorescence microscopy over 3 h using the FITC filter". Where exactly was fluorescence measured? In the entire chamber? In the microchannels? In the outlet? Perhaps indicate in the figure of the chamber the points of analysis.

Fluorescence was measured in a rectangular region of interest, in the inoculation chamber close to the opening of microchannels. This has been added to the text and in the legend of figure 2.

c) Line 76: Inform the composition of the mineral medium or cite a reference to the composition.

The detailed composition of MMG medium has been added to paragraph 2.4

d) Section 2.5: What medium was used? What temperature?

Experiments were performed in MMG medium at 22°C. This is now explained in section 2.5

e) Section 2.5: "maximum and minimum diameter of the colony were measured using ImageJ". For each image, only 1 value of maximum diameter and 1 value of minimum diameter were taken? Why was the median value used instead of the average?

Measurements were performed manually since the contrast was low and the colonies almost perfectly circular. The computed value is the average between minimum and maximum (which is also a median value, given the number of points). The sentence has been changed.

f) Section 2.5: In "Time-averaged elongation velocities were calculated as", include the symbol v_m .

The sentence has been modified.

g) All image analysis was performed in ImageJ? This is not clearly stated.

All image analysis was indeed performed with ImageJ. We modified the text in section 2.5 to clarify this.

6) Results

a) What are the hyphal diameters of the strains tested when grown without confinement? Figure 1b indicates that *N. crassa* is larger than the microchannels. How does it affect growth? These information are given in the discussion but should be presented here.

Comments about the hyphal diameters of *N. crassa* and *T. helicus* have been moved to the Results section.

b) The data presented in Figure 2 was measured in the inoculation chamber or the microchannels? What does the position y refer to? It would be better to indicate the y positions in the figure of the device.

We added a reference to figure 1(a), where position y is defined.

c) Line 103: The radius of the perfusion chamber is 1 mm, which is the length the authors used to calculate the characteristic time. The experimental time is shorter, which is expected if the measurement was done in a radius lower than 1 mm and if convective flow occurs in the chamber.

The characteristic time was computed based on the length of the microchannel, not the width of the chamber. The text has been modified to better explain this choice.

d) Line 106: "Figure 2(a) reveals that the concentration in the inoculation chamber increased fast in the first 30 min, and reached a plateau for the next two hours." According to Fig 2a, it actually increased fast in the first 10 min and, based on fig 2b, it reached a plateau in 30 min (the curves in fig 2b overlap after 30 min).

The referee is right, we modified the description of this figure.

e) Line 109: "This axial dependency can be explained by convective transport of the fluorescein solution through the microchannel." This sentence is misleading. The authors refer to axial dependency in the chamber caused by flow in the microchannel, but it seems that it refers to axial dependency in the microchannel. The authors need to improve the added text in lines 109 to 113, because it is only understandable after reading the appendix.

The text has been modified. It now summarizes the arguments that are developed in more detail in the Appendix and should be understandable by itself.

f) Line 122: How does inoculum size affect the velocity of individual hyphae? The metabolic state hypothesis might be valid, in addition to compounds from the media diffusing to the perfusion, but the inoculum size does not affect individual hyphae.

The solid inoculum contains both mycelium and nutrient-rich agar in variable proportions, which is likely to cause inter-microsystem variations.

7) Discussion:

a) Line 178: "When all the microchannels contain hyphae, we expect their permeability, and as a consequence the gradient in the inoculation chamber, to vanish" Do you mean the fluid will not reach the inoculation chamber when hyphae are present in the channels? This may be true for *N. crassa*, which seems to occupy the entire channel width, but not for smaller hyphae.

Although *T. helicus* hyphae are thinner, the fungus tends to grow until channels are clogged (either by one hypha for *N. crassa* or by a bundle of hyphae for *T. helicus*). The sentence has however been attenuated to take into account the situation of channels that are only partially obstructed by a single *T. helicus* hypha.

b) "The fact that hyphal elongation velocity does not depend on the channel position y , even when some microchannels remain empty, seems to suggest that this situation occurs even before all the channels are filled with hyphae." I don't think you can conclude this. Since there is no difference in extension rate between the static and the perfusion experiment, I believe the nutrients are not being depleted in the static or perfusion experiment. In fact, the medium used in the perfusion does not contain a carbon source, which is the nutrient whose lack of limits growth. The nutrients present in the mineral medium would unlikely be depleted because only small amounts of them are used. Therefore the only effect of the flow would be frictional or removal of biological compounds related to quorum sensing or the mineral medium is washing out the nutrients from the inoculum. In order for you to conclude that there is no flow in the chambers, you should have replaced the circulating buffer with fluorescein after the hyphae had occupied the channels. The nutrients in mineral media are not enough for growth.

For *T. helicus*, there is indeed no difference between static and dynamic experiments, but there is one for *N. crassa*. This suggests that for this strain, increasing the nutrient flux increases the elongation velocity. The fact that the elongation velocity is roughly the same in all microchannels is consistent with the hypothesis that the presence of hyphae in the channels block the flow and that the incoming flux is almost purely diffusive, and similar in all channels of the systems. We rewrote this paragraph and hope that the discussion gained clarity.

- c) Line 200: "Variations in elongation velocities between experimental conditions could be explained by a modification of nutrient availability". Which experimental conditions? Which variations in elongation?
The sentence was indeed unclear and has been rephrased, with a reference to the figure 6b that illustrates the aforementioned variations.
- d) Line 211: There is a discussion about glucose, but there is no glucose in the medium used. Hyphae are either using carbon from the agar of the inoculum or intracellular reserves.
This was a misunderstanding due to our incomplete description of the medium, which has now been modified. The MMG medium does contain glucose, which is the reason why glucose transport is discussed in the manuscript.
- e) Line 217: it is also limited by the lack of space.
We agree with the referee, and the sentence has been modified accordingly.
- f) "It thus seems that the growth of *N. crassa* is limited by the flux of nutrients diffusing from the perfusion chamber, in static and probably also in dynamic conditions". But there is no carbon source in the perfusion chamber.
This was again a misunderstanding due to our incomplete description of the medium, which has now been modified. In dynamic conditions, the medium flow generates a constant influx of glucose, which is a carbon source.
- g) "Its metabolic rate may already be maximal given the geometric constraints." I don't understand this sentence.
This sentence was indeed unclear and has been replaced.
- h) Line 224: "Differences in nutrient consumption rate between the two strains might explain why *N. crassa* is more affected than *T. helicus* by nutrient deprivation in microfluidic conditions". I disagree given what is written here. The probable explanation is lack of space, since the diameter of *N. crassa* is larger than the size of the channel.
For *T. helicus*, increasing the nutrient flux by adding a perfusion does not modify the elongation velocity. This seems to indicate that the lack of space is indeed the only reason of its slow growth. It is true that *N. crassa* is even more affected by the lack of space. However this lack of space is the same whether medium is flowing or not, and the elongation velocity varies between these two conditions. This suggests that both lack of space and nutrient deprivation slow down the growth of *N. crassa* in microsystems under static conditions.
- i) Line 225: "elongation velocity of both strains is so small in comparison with the growth velocity measured at the scale of the colony". The authors can't compare growth in agar with confined growth. In agar, there is substrate (carbon source) available under the colony and O₂, which are lacking in the device. In addition, there are walls blocking hyphal growth in the device.
There are indeed many parameters that differ between the microsystem and Petri dish experiment. We modified the sentence to draw a more explicit list.
- j) Figure 7 should be in the results section.
The figure and the corresponding description have been moved to the Results section.
- k) You could check if it is the wall effect by measuring linear extension rate after the channels and comparing.
The referee is right. However, as hyphae grow out of the microchannels, it becomes more difficult to track them. Individual measurements of hyphal tip velocity would require more sophisticated image analysis, for example in order to differentiate between branching and intersecting hyphae. Also, the time scale used for assessing the instantaneous velocity in microchannels may not be adapted for 2D growth, because events such as branching or encounter between two hyphae are more frequent.
- 8) Conclusion:
a) Line 280: "This suggests that *N. crassa*, although widely used in the literature, may not be the best experimental model for studies related to soil." I don't understand how you can conclude this.
What we meant in this sentence is that *N. crassa* seems very sensitive to confinement, which is a condition encountered in soils. So results obtained in lab-scale experiments in unconfined solid or liquid culture should be handled with care when extrapolating to soils.
- 9) Appendix:
a) What does y represent? Is it position in the microchannels? Please cite the corresponding figure.
We added a reference to figure A1, where position y is defined.
b) Figure 8 should be figure A1 because it is in the appendix. Table should be numbered A1.
The numbering of figures and table in the appendix has been modified.
c) Line 317: Pressure drop refers to a loss in fluid pressure between 2 points. In "pressure drop in the inoculation chamber", what are the two points? In "pressure drop in the perfusion chamber", what are the two points?

The pressure in a microfluidic chamber is usually constant in a cross-section and only depends on the y-coordinate. This has been added to the text.

d) Line 318: write constant instead of cst.

The expression has been modified.

e) Line 331 “This is in agreement with the assumptions made at the beginning of the calculation (unperturbed perfusion how, negligible pressure drop in the inoculation chamber).” Ending the appendix with this statement gives the wrong idea that the assumptions used as starting point are only confirmed by the deduction made, which is not entirely true and which does not prove that the assumptions are true because they were input to the deduction. Whenever possible, assumptions should be proved in the middle of the deduction (e.g. that flow in the channels is insignificant compared to flow in the large chamber).

The text of the Appendix has been modified to better highlight the conclusions that can be drawn from the calculation.

Appendix E

Paris, May 6th, 2020

Dear Editor,

Please find enclosed a third revision of our manuscript, where the modifications have been highlighted in color, as well as a response to the reviewer's comments. We thank you sincerely for your support and hope that you will now find our manuscript suitable for publication in *Royal Society Open Science*.

On behalf of the co-authors,

Anne Le Goff

--

Reviewer: 3

Comments to the Author(s)

The present manuscript presents a study on the extension rate of individual hyphae in a microfluidic device. The objective of the work is clearer in this version and all proposed modifications were included in the new version.

I have a few remaining concerns listed below. The main concern is the characterization of mass transport inside the chambers, reported in Figure 6. This data needs to be further analyzed in order to conclude that convection does occur. It seems that the time required to reach steady-state is too long for the characteristic time calculated by the authors, therefore, they should include other analysis.

I assume that the reviewer is referring to Figure 2, as Figure 6 has no direct link with mass transfer. We addressed the comments on both figures. The modifications are explained in the list below.

A detailed list of suggestions is presented below:

1) Abstract: "This study is one of the first to consider the degree of confinement within soil microporosities as a key factor of fungal soil colonization along with physicochemical parameters, notably for bioremediation purposes." Is it really the first? In the introduction, it is stated "By using micro-patterned surfaces to probe the growth of *Candida albicans* in filamentous form, geometric constraints have been shown to affect hyphal growth and morphology [13]." So there have been other works considering the degree of confinement.

As the referee points out, a few studies in the literature, including references [13] and [18], involve the growth of fungi in confined microenvironment. To the best of our knowledge, our work is the first to combine this approach with a perfusion system allowing to tune water and nutrient transport, with a perspective of developing tools in the field of bioremediation.

Also, what are the sizes of pores in the soil microporosities? In the discussion, there should be a detailed argument to this sentence from the abstract.

The parallel microchannels have a size that is comparable with that of pores in, or between, soil aggregates. A sentence and a reference have been added to the conclusion.

2) Figures:

a) Figure 2: review punctuation in the legend

The legend has been modified

b) Figure 4: Was it the same experiment?

The data plotted in the figure correspond to four separate experiments. The two experiments with *T. helicus* were performed simultaneously, while data corresponding to *N. crassa* were obtained on two different days.

c) Figure 5b: “In 4 of the channels observed,” It is not clear if it is for static experiments or all experiments. It’s better to state “In some channels,”
This sentence has been modified

d) Figure 6: “In both static and dynamic conditions, the difference between *N. crassa* and *T. helicus* was significant ($p < 0.001$).” Based on the error bars, for the dynamic experiment, there doesn’t seem to be a significant difference between both fungi; also, for the same fungus, there aren’t significant differences between the treatments. The only significant difference that may occur is between both fungi in static conditions. Please, review this analysis. In addition, v_c is explained twice in the legend, it is only necessary in the first appearance, using ().

For clarity, we have chosen to express all quantities as mean \pm standard deviation throughout the manuscript. Accordingly, error bars in the figures represent standard deviation. Data plotted in figure 6 correspond to a large number of individual measurements, which explains why the difference between samples can be significant in spite of overlapping error bars. If we use standard error (SEM) instead of standard deviation to plot error bars, as is often done with large samples, the effect is more obvious, as illustrated in the figure below.

The legend has been edited according to the comment.

3) Introduction:

a) “The objective is to be able to discriminate between phenomena that result from the geometrical constraints and those that are merely a consequence of starvation” How do you achieve this?

The sentence has been modified to explain that the methodology is based on the comparison between static and dynamic experiments. The detailed reasoning is explained in the Results/Discussion

2) Experimental section:

a) “Strains were maintained on malt yeast extract agar medium (MYEA)”. Please inform the composition of this medium, since there are variations in the literature.

The composition of MYEA medium has been added in the Experimental section.

b) Section 2.5: It is written that the plate experiment was done in MYEA. In the reply, it is stated that it is MMG. Verify this information. If the medium used in the chamber experiment and in the plate experiments were different, how can you compare their growth?

Microfluidic experiments were performed in MMG This medium is known to be suited for fungal growth [3]. Its composition does not involve biological ingredients, which allows for a fine control of nutrient uptake. Plate experiments were performed in MYEA. We also performed experiments on solid MMG and obtained similar trends, but we found it more interesting to compare microfluidic growth speed with data widely available in the literature, as rich media such as MYEA are routinely used to assess fungal growth [16, 18, 19]. There is of course no reason why the growth velocities

should be the same in micro and macro experiments. Many experimental conditions differ (not only the medium composition). We are merely pursuing the goal of trying to draw correlations between observations made at different scales.

3) Results

a) “The diameter of *N. crassa* hyphae is about 10 μm , in contrast to those of *T. helicus* which are 2 to 5 μm wide depending on their age” Avoid the use of “those”, it is better to state “in contrast to the diameter of *T. helicus* which is 2 to 5 μm , depending on their age”. Also, with this information, you should mention the size of the channels again, reflecting how hyphal diameter affects growth in the channels.

The sentence has been modified

b) The authors use the characteristic time of 10 min, which is the time necessary for fluorescein to be detected in $y=0$, for the validation that convection occurs. However, this is a 1 measurement data, which may have errors in measurement. The authors should use the rest of the data in figure 2 to validate this affirmation. It could be done with mathematical models or equations for the diffusion profile (bell-shape graphs). The profile in figure 2b is unusual because the bell-shape graph should become flat over time, which does not occur. Please, improve the analysis of the data in figure 2 in order to confirm the transport phenomena occurring in your device.

We indeed referred to the curve at $y=0$ to estimate the time scale of variations in fluorescence intensity because the amplitude of variations is largest on this curve. We checked however that this particular choice does not have a strong impact on the characteristic time. If we normalize, for each position, the fluorescence intensity by its value at 150 minutes, data collapse onto a single curve, demonstrating that the characteristic time does not vary with y .

The reviewer was expecting a bell-shaped curve instead of the curve shown in Fig 2 (a). Such curves would indeed be observed if a pulse of dye was injected in the microchannel. These experiments aim at quantifying dispersion in fluid networks. In our case, we rather wanted to mimick the implementation of a continuous nutritive flow in the system. The fluorescein concentration thus increases abruptly from zero to a finite value and remains constant throughout the perfusion, which explains why the intensity reaches a plateau instead of decreasing.

We modified the sentence in the text, to better explain that the 10 minutes we are referring to represent the delay seen in figure 2 (a) or in the above graph. It is true that the very shape of the curve could be, as suggested by the reviewer, modeled by coupling advection and diffusion using the flow field established in the Appendix. We chose to not develop that part, as our main message regarding the flow in parallel channels, is that, although our microsystem looks like a gradient generator and can function like one in the absence of fungi, the flow that fuels the gradient is suppressed when channels are occupied by hyphae. In the rest of the manuscript, we therefore consider that all parallel microchannels are equivalent.

c) Line 155: Inoculum size refers to the amount of mycelium in the inoculum, which does not affect the extension rate of individual hyphae, only the number of hyphae formed. If the authors mean that the amount of nutrient-rich agar is affecting growth, than they have to replace “inoculum size” for “amount of nutrient-rich agar added with the inoculum”.

As we know that the amount of nutrients is a factor determining the growth speed, variations in the amount of nutrient-rich agar is an obvious source of variability. As for the amount of mycelium, its influence is less clear. Since we have no argument to rule out its role, we decided to keep it in the list of potential sources of variability. The sentence has been edited for clarity.

Appendix F

29th July 2020

Dear Editor,

We are grateful to the referees and the Editorial Office for their help in improving the manuscript and we thank you very much for accepting our manuscript for publication in *Royal Society Open Science*.

Please find a final version of our manuscript taking into account the last modifications suggested by the referee.

Sincerely yours,

Anne Le Goff, on behalf of the coauthors

Anne Le Goff
Assistant Professor

Department of Bioengineering

+333 44 23 79 55
anne.le-goff@utc.fr

**Université de technologie
de Compiègne**

Rue du Docteur Schweitzer
CS 60319
60203 Compiègne cedex

Tél. 03.44.23.44.23
www.utc.fr